# Comparative Metabolomic Approaches to Nanoplastic Toxicity in Mammalian and Aquatic Systems

**DOI:** 10.3390/ijms27010050

**Published:** 2025-12-20

**Authors:** Junhyuk Lee, Hyeonji Jang, Boyun Kim, Jewon Jung

**Affiliations:** 1College of Fisheries Science, Pukyong National University, Busan 48513, Republic of Korea; arzi777@pukyong.ac.kr; 2Department of SmartBio, College of Life and Health Science, Kyungsung University, Busan 48434, Republic of Korea; hyeonji00201@gmail.com

**Keywords:** nanoplastics, nanotoxicity, metabolism, metabolomics, lipidomics

## Abstract

Nanoplastics (NPs), emerging contaminants originating from the degradation of larger plastics, have raised significant environmental and health concerns due to their ability to penetrate biological barriers and disturb cellular homeostasis. Exposure to NPs has been shown to induce oxidative stress, mitochondrial dysfunction, and inflammatory responses in both mammalian and aquatic systems, ultimately leading to metabolic imbalance. Metabolomics, a comprehensive analytical approach focusing on small-molecule metabolites, provides a direct reflection of these biochemical alterations and offers critical insights into the mechanisms underlying NP-induced toxicity. This review summarizes recent metabolomic studies investigating nanoplastic toxicity across mammalian and aquatic organisms, highlighting commonly perturbed pathways such as lipid metabolism, arachidonic acid metabolism, the tricarboxylic acid (TCA) cycle, and amino acid metabolism. These disruptions indicate that NPs impair energy production, lipid regulation, and redox balance. In mammals, polystyrene and polyethylene terephthalate nanoplastics have been shown to alter hepatic and intestinal metabolism and induce oxidative and inflammatory stress, while in aquatic species, similar metabolic disturbances occur in the gills, liver, and brain. Collectively, the evidence emphasizes metabolomics as a powerful approach for elucidating the molecular basis of nanoplastic toxicity and suggests that integration with other omics techniques is essential for comprehensive risk assessment and mechanistic understanding.

## 1. Introduction

Plastics are widely used in various industries, with the most commonly used, accounting for approximately 90% of global production, being polyethylene (PE), polypropylene (PP), polyvinyl chloride (PVC), and polystyrene (PS), respectively [1,2,3]. Environmental contamination from plastic waste is now evident worldwide, and smaller fragments are categorized as microplastics (MPs) or nanoplastics (NPs). According to the European Chemicals Agency, MPs are defined as particles ranging from 100 nm to 5 mm, while the U.S. National Nanotechnology Initiative defines NPs as particles between 1 and 100 nm [4,5]. However, in many studies, the size of nanoplastics is generally defined as particles within the size range of 1 nm to 1 μm. This results in a partial overlap between the two categories, and there is some debate on the exact boundary [6]. Microplastics can be fragmented into nanoplastics through physicochemical stress, while nanoplastics may aggregate under environmental conditions to form larger assemblies, and therefore, the two types of particles are closely related [4,7]. However, the two types differ in characteristics and biological consequences, and should be considered distinct [5]. Because the size definition of nanoplastics differs among studies, the particle-size ranges applied in the literature are not always consistent. In this review, we adopt the 1 nm–1 μm range for the term ‘nanoplastics,’ as this classification aligns with the definitions most commonly used in metabolomics-based research. It is important to note that despite the production quantities, nanoplastic toxicity studies are leaned towards polystyrene [8]. Additionally, most laboratory toxicity studies use engineered spherical nanoplastics, which differ from those monitored in real environmental settings. So the resulting data may not fully represent plastics to which organisms are actually exposed [9]. Nanoplastics introduced into aquatic environments can accumulate in resident organisms through various exposure routes, such as the gills, epidermis, and intestine [10], and have indeed been detected in a wide range of organisms, including fish, mammals, and crustaceans [11,12,13,14]. Humans, on average, ingest approximately 11,000 microplastic and nanoplastic particles per year through the consumption of seafood such as oysters, crabs, and fish [15]. Exposure to nanoplastics via drinking water and inhalation of water/airborne particles and fibers is also possible [7,14,15], and particles have been identified in human stool and urine samples [16].

Given their ability to penetrate biological barriers and interact with intracellular organelles, nanoplastics elicit a variety of toxic responses at both the molecular and cellular levels. A major outcome of nanoplastic exposure is the excessive generation of reactive oxygen species (ROS), which depends on particle size, surface properties, and exposure conditions [17,18]. It has been demonstrated that nanoplastics such as polystyrene (PS), polyethylene terephthalate (PET), and polypropylene (PP) elevate intracellular ROS levels, not only at the tissue level but also at the cellular level [19,20,21,22,23]. Specifically, ROS produced following nanoplastic exposure have been shown to impair mitochondrial function, leading to reduced adenosine triphosphate (ATP) synthesis and disruption of the tricarboxylic acid (TCA) cycle [19,23,24,25,26]. Nanoplastics have also been shown to elevate mitochondrial ROS (mROS) production in various cell types, leading to mitochondrial membrane potential collapse and mitochondrial damage. Moreover, nanoplastic exposure significantly decreases ATP synthesis, thereby causing mitochondrial dysfunction [23,26]. Such alterations are not limited to the cellular level but are also evident at the tissue level, where mitochondrial swelling and structural deformation have been observed. These effects are accompanied by ROS overproduction and oxidative stress, which markedly alter cellular morphology [19]. Excessive production of ROS activates multiple intracellular signaling pathways, including the mitogen-activated protein kinase (MAPK) pathway [27]. This pathway regulates various stress responses such as inflammation, autophagy, and apoptosis [28,29]. Consequently, nanoplastics can act as particulate xenobiotics that provoke immune activation, as evidenced by increased levels of pro-inflammatory cytokines and chemokines and enhanced infiltration of immune cells in both tissues and cultured systems [23,30]. Furthermore, nanoplastics have been reported to induce cyst formation containing purulent material and to cause inflammatory lesions [30]. Polystyrene nanoplastics (PS-NPs) also increase lipid peroxide accumulation and malondialdehyde (MDA) levels at the cellular level and form lipid droplets. They also disrupt lipid metabolic pathways, thus destroying lipid homeostasis and causing cellular stress and pathological changes [19,30,31,32]. Collectively, these findings highlight that nanoplastic exposure triggers a cascade of oxidative stress, mitochondrial dysfunction, and inflammatory signaling that converge on metabolic imbalance. Because these toxic responses ultimately manifest as changes in cellular metabolism, analyzing alterations in metabolites provides an effective window into the underlying mechanisms of nanoplastic toxicity. In this context, metabolomics has emerged as a powerful approach to systematically capture and interpret these biochemical perturbations.

Metabolomics, the comprehensive study of small-molecule metabolites that represent the end products of cellular metabolism, is a key branch within the broader field of omics and bioinformatics. Omics refers to the integrated technologies of genomics, transcriptomics, proteomics, and metabolomics, and is used to explore the structures, functions, relationships, and dynamics of biomolecules within an organism [33]. It starts from the idea that information from the genes is transferred to the phenotype through the route of gene-transcription (mRNA)–protein (enzyme)–metabolite. However, biological systems are governed by intricate feedback mechanisms and regulatory networks; therefore, gene expression alone does not fully account for cellular responses [34,35]. As demonstrated above, while the expression of genes affects the phenotype, it does not mean that all genes are translated into functional products. There is a divergence in the scale of transcription, protein turnover, and metabolite production, and some are even decoupled from gene expression. Metabolomics aims to investigate these chemical processes (so-called metabolic pathways) by using end products (metabolites) and by obtaining these metabolite profiles. Metabolites are highly sensitive to exogenous stimuli and serve as direct indicators of environmental and toxicological stress [36]. This is effective in understanding the toxicity of materials, which is why metabolomics is an excellent method for environmental pollution investigation and can progress towards a concrete understanding of the biological mechanisms with the integration of other omics methods. It is also advantageous due to the fact that the majority of metabolite structures are not species-specific, while many nucleic acid and amino acid sequences of genes and proteins are [37]. In order to detect these metabolites, researchers use innovative methods such as nuclear magnetic resonance spectroscopy (NMR), Liquid chromatography–mass spectrometry (LC-MS), gas chromatography–mass spectrometry (GC-MS), etc. [38]. While the detailed execution methods differ, the basic concept is to identify each metabolite by its distinct characteristics. Although metabolomics offers clear advantages for characterizing biochemical alterations, several methodological limitations must also be acknowledged. First, metabolite levels are highly sensitive to pre-analytical and analytical conditions, including sampling timing, storage procedures, extraction methods, and variability across analytical platforms (e.g., NMR, LC–MS, GC–MS). Such factors can introduce inter-study variability, particularly in the absence of standardized protocols [39]. Second, accurate interpretation of metabolomic profiles requires detailed knowledge of metabolite chemistry, biological function, and pathway context within each organism, which can complicate cross-study comparisons [39]. Nevertheless, recent efforts toward protocol harmonization, quality-control standardization, and cross-platform validation provide a strong foundation for improving reproducibility and reliability in future metabolomics research [40,41,42]. Therefore, a comprehensive synthesis of current metabolomic findings is essential in order to clarify how nanoplastic exposure alters key metabolic pathways across different biological systems. This review summarizes recent metabolomic studies investigating nanoplastic toxicity in mammals and aquatic organisms, emphasizing disrupted pathways such as energy, lipid, and amino acid metabolism. By integrating these results, we aim to identify common biochemical signatures and propose potential mechanistic links between nanoplastic exposure and metabolic dysregulation.

## 2. Altered Metabolism and Toxicity Induced by Nanoplastic Exposure

Nanoplastics are exposed to various animal species through various environmental media, and the main routes of exposure in the human body are known as oral ingestion and inhalation [11,12,15,43,44]. Nanoplastics introduced into the body move along the blood circulation system and can cause functional and pathological damage to several organs [21,31,45]. The orally exposed nanoplastics are absorbed through the intestinal epithelium and then distributed throughout the body through the bloodstream and lymphatic systems [46,47]. Inhaled nanoplastics enter the blood through the alveolar barrier and then reach various organs through the circulatory system [47,48]. During this circulation process, nanoplastics may pass through biological barriers such as the blood–brain barrier (BBB) or placenta due to their microscopic size and accumulate in various tissues [49,50]. Accumulated nanoplastics mainly induce the production and inflammatory reactions of ROS, resulting in cell damage and decreased tissue function [51,52,53]. Several studies have shown that the intensity and pattern of these toxic reactions differ depending on the type, size, surface characteristics, and type of organ to be accumulated [22,30,54].

This section focuses on representative metabolomic alterations and key pathway-level alterations commonly identified across mammalian species. An overview of the metabolic pathways and organ-specific responses is shown in Figure 1, providing a visual summary of nanoplastic-induced metabolic reprogramming in mammals, and the altered metabolic pathways identified in each study are summarized in Table 1. Detailed datasets and species-specific responses are available in the cited studies. Most of the metabolic studies covered in this section used a UPLC-MS-based analysis, and only applications of other analysis techniques will be mentioned.

### 2.1. Tissue-Specific Alterations in Metabolism and Toxicity

#### 2.1.1. Intestinal Toxicity and Metabolic Disruption Induced by Nanoplastics

Nanoplastics are absorbed through the intestinal barrier and then transported through the systemic circulatory system to various organs, such as the liver [55]. This process is highly dependent on the size of the particles and the surface charge, and the smaller the size and the more negatively charged the surface, the higher the absorption rate is [56]. The gastrointestinal tract (especially the small intestine and colon) is the main organ by which nanoplastics may be first absorbed and affected in the human body, and the absorbed nanoplastics cause an imbalance in the intestinal microbial environment, destroying intestinal homeostasis. It has also been reported in several studies that this induces pathological changes, such as reducing intestinal mucus secretion and increasing the likelihood of cyst formation and tumor formation [30,53]. These results suggest that nanoplastics may impair the structural and immunological stability of the intestinal barrier, contributing to the deterioration of intestinal function and the development of inflammatory diseases.

Metabolite analysis was mainly performed in the intestine and liver, reflecting that both organs are major targets for nanoplastic toxicity [55,57,58]. In studies in which PS-NPs were exposed to Caco-2 cells, retinol metabolism, drug metabolism by cytochrome P450, and arachidonic acid metabolism pathways were disturbed. In addition, the expression level of the transcriptional regulator SREBP (Sterol Regulatory Element-Binding Proteins) was significantly increased, which led to abnormalities in fatty acid composition during chronic exposure. Under long-term exposure conditions, PS-NPs disrupted various steroid-related metabolic pathways, including fatty acid degradation/elongation, terpenoid backbone biosynthesis, steroid hormone biosynthesis, and steroid biosynthesis [54].

In AOM/DSS-treated mice, PS-NP exposure altered several intestinal metabolites, including arachidic acid and stearic acid, which are closely linked to fatty acid metabolism and NF-κB-mediated inflammatory signaling, and these changes were confirmed by LC-MS analysis [19]. Similarly, PS-NPs also induced significant changes at the cellular level. In lipopolysaccharide (LPS)-treated Caco-2, glucose metabolism was promoted, lipid peroxidation was increased, and fatty acid metabolism was suppressed, and these metabolic changes were also identified by LC-MS-based analysis [19]. In addition, exposure to PS-NPs significantly enriches the ABC transporter (ATP-binding cassette transporters), propane/piperidine/pyridine alkaloid biosynthesis, phenylpropanoid biosynthesis, and bile secretion pathways. These changes led to disturbances in bile- and lipid-related metabolic pathways, and as a result, mucus secretion was significantly reduced [53]. In addition, major amino acid metabolites such as tyrosine, proline, glutamine, tryptophan, and glutamate changed in neurotransmitter-related pathways, and the fluctuations in these metabolites were significantly correlated with an anxiety-like behavior index and gut permeability [53].

Collectively, these findings indicate that PS-NPs perturb multiple metabolic processes—particularly lipid, arachidonic acid, and energy metabolism—which are central to cellular homeostasis. Arachidonic acid is an omega-6 fatty acid derived from cell membrane phospholipids and is a major metabolic precursor in the inflammatory response. When arachidonic acid metabolism is activated, it promotes inflammation, oxidative damage, and lipid metabolic disorders [59]. The concurrent activation of lipid and energy metabolism enhances membrane remodeling, inflammatory signaling, and lipid peroxidation, ultimately causing mitochondrial overload, ROS accumulation, and cellular injury [60]. Taken together, metabolomic data from mammalian models reveal a consistent mechanistic pattern linking nanoplastic exposure to oxidative stress, lipid imbalance, and inflammation-driven metabolic dysfunction.

#### 2.1.2. Hepatic Toxicity and Metabolic Disruption Induced by Nanoplastics

The liver is directly connected to the intestine via the portal vein, allowing nanoplastics absorbed from the gut to be transported to the liver through portal circulation [56,61]. Disruption of the gut–liver axis balance can directly influence hepatic metabolism and disease development. As one of the body’s largest digestive organs, the liver plays a central role in detoxification and metabolic regulation [62]. Therefore, accumulation of nanoplastics in the liver can impair detoxification capacity and cause metabolic imbalances [63]. Nanoplastics accumulated in the liver cause hepatocellular damage, an inflammatory response, and pathological damage. Nanoplastics exposed to the liver through the intestine may interfere with liver metabolism and immune homeostasis, resulting in potential toxicity. Studies that exposed PS-NPs to mouse models have significantly altered glycerophospholipid metabolism, linoleic acid metabolism, arachidonic acid metabolism, glyceride metabolism, and fatty acid metabolic pathways, and these metabolic changes were also identified by UPLC-MS and NMR-based analysis [64,65]. In particular, most glycerophospholipid metabolites were downregulated, and disturbances in arachidonic acid and glycerophospholipid metabolic pathways were noticeable [64]. In human hepatocyte (L02) studies, PS-NP exposure affected nicotinamide adenine dinucleotide metabolism (NAD^+^/NADH metabolism) and caused substantial dysfunction in the mitochondrial urea cycle and electron transport chain. These processes are closely associated with the TCA cycle, glutathione (GSH) metabolism, and purine metabolism, indicating PS-NP–induced mitochondrial impairment. Notably, PS-NPs increased malate levels while reducing fumarate levels, resulting in TCA cycle imbalance [26]. In addition, in a study that exposed polycarbonate nanoplastics to a human hepatocellular model (UHHS), the expression of CYP3A4 and CYP2C9 among cytochrome P450 (CYP) enzymes and enzyme activity were suppressed, and the expression of albumin genes was also decreased. These results imply a decreased metabolic and protein synthesis function of liver cells, and these metabolic changes were also identified by LC-MS-based analysis [51].

In a study in which polyethylene terephthalate nanoplastics (PET-NPs) were exposed to mouse models, small-sized PET-NPs resulted in the upregulation of lipid metabolites (Cholesterol, 4,4-dimethylcholest-8(9),24-dien-3β-ol, cholesta-5,24-dien-3β-ol, 22β-hydroxycholesterol, glycocholic acid, propionic acid, nicotinamide, ectoine, xanthine) and a reduction in arachidonic acid, anserine, and histamine. On the other hand, in the case of large PET-NPs, they induced the downregulation of sphinganeine. These metabolomic alterations indicate the abnormal activation of lipid, energy, and amino acid metabolism pathways, ultimately leading to cellular stress and pathological changes [30]. Exposure to various nanoplastics causes mitochondrial dysfunction, lipid and energy metabolism disturbance, oxidative stress, and inflammatory reactions in liver tissue. These metabolic disorders lead to the loss of metabolic homeostasis and protein synthesis ability in hepatocytes, and they cause liver dysfunction and tissue damage.

#### 2.1.3. Lung Toxicity and Metabolic Disruption Induced by Nanoplastics

Nanoplastics are introduced into the lungs in the form of inhalation exposure through airborne particles such as masks and fine dust [23,48]. Inhaled nanoplastics enter the body through the alveoli and bronchial epithelium, and some move to the circulatory system through lung capillaries or lymphatic systems [66]. Nanoplastics diffused throughout the body cause oxidative stress in cells and promote inflammatory response and apoptosis to exhibit toxic reactions [23,26,67]. These results suggest that respiratory exposure may cause potential toxicity not only to the lungs but also to whole-body tissues. Studies of the exposure of PS-NPs to pulmonary epithelial cells (BEAS-2B cells) have identified disturbances in arginine biosynthesis and the alanine, aspartate, and glutamate metabolism pathways. Notably, clear abnormalities in the urea cycle and electron transport chain in mitochondria were observed, indicating that PS-NPs induce mitochondrial dysfunction. These metabolic disturbances were closely related to the TCA circuit, GSH metabolism, and purine metabolism, resulting in the collapse of the intracellular energy metabolic imbalance and redox balance. In addition, the concentration of citric acid was significantly reduced, and the concentration of fumarate increased, which is interpreted to promote the accumulation of mROS, causing the imbalance of the electron transport system and the increase in oxidative stress [26].

Meanwhile, studies that exposed PET-NPs to mouse models have shown increased L-alanine biosynthesis, glyoxylate degradation, D-galactate degradation, and L-arginine metabolism pathways in lung tissue, and these metabolic changes were also identified by GC-MS-based analysis [68]. These metabolic changes collectively suggest that nanoplastic exposure induces reorganization of the amino acid metabolism and energy production pathways. Overall, exposure to PS- and PET-based nanoplastics consistently induces mitochondrial dysfunction, energy metabolism impairment, and amino acid metabolic disturbance in the lungs. These alterations act synergistically to promote oxidative stress and inflammatory reactions, establishing a common mechanism of lung cell injury under nanoplastic exposure.

#### 2.1.4. Brain Toxicity and Metabolic Alterations Induced by Nanoplastics

The brain plays a central role in coordinating body function [69,70]. Nanoplastics can reach the central nervous system (CNS) through multiple exposure routes, including systemic circulation and inhalation. Due to their small size and hydrophobicity, micro- and nanoplastics can bypass the blood–brain barrier (BBB), where they may induce neuronal cytotoxicity, disrupt molecular pathways, and potentially contribute to neurodegenerative disease progression [71,72,73]. The deposition of ultrafine particles onto the olfactory epithelium also enables direct “nose-to-brain” transport via the olfactory nerve [73]. Once nanoparticles enter the brain, they are recognized by microglia, triggering phagocytosis and neuroinflammatory responses. Recent mouse studies demonstrate that polystyrene nanoplastics increase BBB permeability, activate microglia, and induce neuronal damage [74,75].

Exposure of 5 µm PS-MPs to mice resulted in significant changes in cysteine and methionine metabolism; glycine, serine, and threonine metabolism; sphingolipid metabolism; tyrosine metabolism; and the xenobiotic metabolism regulated by cytochrome P450 in the frontal lobe of mice [76]. Mixed exposures to PS, PE, and PLGA similarly altered glycerolipid metabolism, steroid biosynthesis, tyrosine metabolism, and xenobiotic metabolism. Low-dose exposure upregulated metabolites related to D-glutamine and nitrogen metabolism, whereas high-dose exposure enhanced the valine, leucine, and isoleucine degradation pathways. Dose-dependent disruptions of the kynurenine pathway were observed, with low-dose PS exposure increasing quinolinic acid and decreasing nicotinic acid and tryptamine. High-dose exposure reduced melatonin and nicotinic acid while increasing quinolinic acid and ADP-ribose. Mixed-plastic exposure increased NADH and reduced L-kynurenine, HIAA, and 3-hydroxykynurenine. Among the altered metabolic pathways, the xenobiotic metabolism regulated by cytochrome P450 occurs in the liver, but changes in the cytochrome P450 pathway in the brain due to MP exposure suggest potential neurotoxic effects [76]. Oxidized PS-NPs (ox-PS-NPs) induced significant elevations in multiple amino acids—including alanine, leucine, isoleucine, valine, proline, aspartate, glycine, cysteine, and glutamate—whereas exposure to PS-NP/MP mixtures or oxidized PS-MPs resulted in noticeable depletion of the same metabolites. ox-PS-NPs significantly increased ATP/ADP, glucose, malate, and malonate concentrations, but the opposite trend was observed when exposed to ox-PS-MPs and PS-NPs/PS-MPs. Neurotransmitter-related metabolites, including acetylcholine, choline, acetate, and serotonin, were elevated in ox-PS-NP–treated cells but significantly reduced following ox-PS-MP or PS-NP/MP exposure. The concentration of taurine decreased under all experimental conditions, and the decrease was more pronounced, especially in cells exposed to oxidized ox-PS-NPs, and these metabolic changes were also identified by NMR-based analysis. The rapid increase in glycine, alanine, leucine, isoleucine, valine, glutamine, glutamic acid, aspartic acid, and proline in ox-PS-NPs-treated SH-SY5Y cells suggests that protein synthesis is limited [77]. Glycine, cysteine, and glutamate are known as precursors of glutathione (GSH), and the fact that the concentration of these amino acids has increased suggests that cells need GSH, which plays an important role in relieving oxidative stress. We found a decrease in taurine levels in SH-SY5Y cells exposed to ox-PS-NPs/PS-MPs, which might indicate that the biosynthesis of this metabolite was inhibited. A significant decrease in taurine levels could lead to mitochondrial dysfunction, as confirmed by the observed mitochondrial depolarization [78].

Although multiple studies have examined nanoplastic-induced neurotoxicity, metabolomics-based investigations remain limited. Furthermore, because nanoscale particle data are scarce, microplastic findings were included where mechanistically relevant. This scarcity limits our ability to fully delineate the biochemical basis of nanoplastic-induced brain toxicity. Therefore, additional metabolomics-focused research is required to further characterize nanoparticle-driven metabolic alterations in the brain.

#### 2.1.5. Metabolic Effects in Offspring and Reproductive Organs

Nanoplastics can pass through the blood–testis barrier and move to the spermatogenic region. This causes decreased sperm concentration, decreased motility, increased abnormal sperm rate, and hormonal imbalance [79]. These changes increase the likelihood of developing reproductive dysfunction and can lead to hormonal and endocrine disruption. In addition, nanoplastics can pass through the placenta and be transmitted to the fetus, resulting in physiological changes such as delayed fetal development, decreased average weight, decreased liver size, and absolute liver weight [31]. These results suggest that nanoplastics may exhibit potential toxicity throughout the reproductive and developmental processes.

Metabolomic analyses in the testes of PS-NP–exposed mice revealed increased levels of 3-phospho-D-glycerate, phosphoenolpyruvate, and lactate, together with decreased levels of acetyl-CoA and fumarate, and these metabolic changes were also identified by LC-MS-based analysis [25]. These alterations indicate that metabolic stress caused by PS-NPs disturbs aerobic glycolysis and the TCA cycle, resulting in ATP generation (ATP synthesis inhibition). In addition, PS-NPs increased the expression of metabolites related to DNA damage, such as adenine, adenose, nicotinamide, and uracil in mouse sperm. As a result, the vitamin and nucleoside transport, glycerophospholipid metabolism, nucleic acid metabolism, innate immunity-sensing, base excision repair, and DNA repair-related pathways have been enriched. These changes suggest that exposure to PS-NPs may induce lipid metabolic disturbance, nucleic acid metabolism, and DNA repair function deterioration of sperm cell membranes during pregnancy and lactation [80]. Collectively, these results imply that PS-NPs disrupt lipid and nucleic acid metabolism and impair DNA repair capacity in sperm cells, contributing to reproductive dysfunction.

Maternal exposure during pregnancy and lactation further amplifies these effects, as PS-NPs can cross the placenta and be transferred to offspring via breast milk [25]. As a result, levels of energy metabolites dihydroxyacetone phosphate, β-D-fructose 6-phosphate, adenosine monophosphate, D-glucose 6-phosphate, and thiamine pyrophosphate are significantly reduced in the liver of offspring, and these metabolic changes were also identified by LC-MS-based analysis [31]. These metabolic changes cause glucose metabolism disorder and promote growth delay (weight loss), increased lipid peroxidation, and inflammatory response, which can lead to morphological and functional liver damage [31]. PS-NPs commonly induced decreased energy metabolism and mitochondrial functional disturbance, decreased lipid and nucleic acid metabolism, and increased oxidative stress. These metabolic abnormalities act as major mechanisms mediating decreased reproductive function and delayed offspring development through intracellular energy imbalance and DNA damage accumulation.

#### 2.1.6. Other Metabolic Disorders Due to Nanoplastics

Beyond the organs discussed above, nanoplastics also affect other vital systems, including the kidney and vascular system. Once in the bloodstream, PS-NPs interact directly with erythrocytes and significantly increase intracellular Ca^2+^ levels in a concentration-dependent manner, accompanied by marked depletion of ATP and GSH. PS-NPs also destabilize the erythrocyte membrane by activating scramblase, which disrupts phospholipid asymmetry and promotes microvesicle shedding through phosphatidylserine externalization [81]. These findings indicate hemolytic potential; however, no metabolomics-based studies have yet characterized hemoglobin-derived metabolites or lipid peroxidation markers in erythrocytes, and such data remain unavailable.

Beyond mature erythrocytes, nanoplastics can also impair hematopoietic function. In hematopoietic stem and progenitor cells (HSPCs), PS-NP exposure increases ROS levels in a dose-dependent manner and reduces cell viability. Metabolomic profiling identified eight significantly upregulated metabolites (dimethylglycine, propionic acid, oxoglutaric acid, octanoic acid, nonanoic acid, nonoelaidic acid, 2,2-dimethyladipic acid, and glucose 6-phosphate) and two downregulated metabolites (ethylmethylacetic acid and isocaproic acid). These changes enriched multiple pathways, including the TCA cycle, D-glutamine and D-glutamate metabolism, alanine–aspartate–glutamate metabolism, lysine biosynthesis, vitamin B6 metabolism, propanoate metabolism, and butanoate metabolism. Collectively, these findings suggest that PS-NPs disrupt metabolic homeostasis in HSPCs and may impair hematopoietic capacity [82]. In the kidney—responsible for filtration and excretion—nanoplastic accumulation leads to structural injury, impaired filtration function, and transcriptomic alterations. These changes are accompanied by tubular cell damage and increased oxidative stress, ultimately resulting in functional and morphological abnormalities [83].

Exposure of PS-NPs in human kidney epithelial cells (HKC) downregulated metabolites associated with energy metabolism, such as citric acid, 6-phosphogluconic acid, and pyrophosphate, and upregulated metabolites in the nucleotide metabolic pathway [84]. Changes in glycolysis, the TCA cycle, amino acid metabolism, fatty acid metabolism, and nucleotide metabolism were identified; glutamate metabolism was disrupted; and enzymes involved in lysine degradation, tryptophan metabolism, valine, leucine, and isoleucine degradation were downregulated overall [84]. Upregulation of serine hydroxymethyl-transferase, which catalyzes the interconversion between serine and glycine, was induced, which increases the de novo synthesis of GSH in response to NP-PSs-induced oxidative stress. Fatty acid metabolism, including fatty acid synthase, carnitine O-palmitoytransferase 2, malonyl-CoA-acyl carrier protein transferase, and 3-oxoacyl-[acyl-carrier-protein] synthase, was downregulated, inhibiting amino acid and fatty acid metabolism [84]. The metabolites associated with purine and pyrimidine were increased, which led to an increase in the concentration of metabolites associated with the synthesis of purine and pyrimidine, which are components of DNA, such as deoxyguanosine, deoxyadenosine, and deoxyinosine, leading to an increase in the concentration of uric acid, which is the end product of purine degradation [84]. Corresponding major enzymes, such as hexokinase and alpha-enolase, were downregulated, and electron transfer chains were changed with the TCA circuit, causing mitochondrial energy dysfunction [84].

In mouse models, PS-NP exposure disrupted aminoacyl-tRNA biosynthesis, amino acid biosynthesis, and sphingolipid metabolism, altering the composition of membrane structural lipids such as 1-alkyl-2-acylglycerophosphocholines, diacylglycerophosphocholines, and phosphosphingolipids, and these metabolic changes were also identified by LC-MS-based analysis [85]. Consistently, PS-NP exposure in human hepatoma (Huh-7) cells resulted in extensive metabolic reprogramming, with 297 metabolites differentially expressed—82 upregulated and 215 downregulated—primarily involving phospholipid and glycerophospholipid pathways. Transcriptomic analysis showed consistent results, with PS-NP exposure increasing diacylglycerol kinase expression, promoting the conversion of diacylglycerol to phosphatidic acid. This shift enhanced vesicle formation and reduced triacylglycerol (TG) accumulation within cells, indicating that PS-NPs disrupt intracellular lipid metabolism and affect phospholipid signaling and membrane dynamics [52]. Collectively, these studies demonstrate that nanoplastics destabilize cellular metabolic homeostasis across multiple systems. Common features include the suppression of energy, lipid, and amino acid metabolism; activation of purine–pyrimidine pathways; and compensatory GSH synthesis in response to oxidative stress. Such metabolic reprogramming reflects a pathological adaptation to nanoplastic-induced oxidative and mitochondrial dysfunction, underscoring the systemic nature of nanoplastic toxicity.

**Table 1 ijms-27-00050-t001:** Metabolomic studies on various nanoplastic toxicities in terrestrial organisms (in vivo). Although each study reported distinct metabolic pathways, only the major pathways discussed most prominently are reflected as the key pathways. Asterisks mean that a pathway was commonly reported in multiple studies (* 2 studies, ** more than 3 studies). Brackets in ‘key pathways’ mean either that a metabolite was reported but was not discussed in a specific pathway or that multiple metabolites were described in a vague group.

Nanoparticles	Size (nm)	Concentration	Exposure Time	Model	Method	Sample	Key Pathways	Reference
Polystyrene	60	20 mg/kg/day	35 d	C57 BL/6 J (Mouse)	UPLC-Q -TOF-MS	Testis Sperm	DNA repair pathway Glycerophospholipid metabolism * Innate immune-sensing Nucleotide metabolism * pathway Vitamin and nucleoside transport	[80]
	100	1000 µg/L	28 d	C57 mouse	UPLC-MS	Liver	Arachidonic acid metabolism * Fatty acid metabolism ** Glyceride metabolism Glycerophospholipid metabolism * Linoleic acid metabolism	[64]
	100	0.5 mg/day	60 d	C57BL/6J	UPLC-MS	serum	Alkaloid biosynthesis pathway ATP-binding cassette (ABC) Bile secretion pathway Phenylpropanoid biosynthesis transporters pathway Tropane, Piperidine, and Pyridine pathway	[53]
Polystyrene	80	10, 100 µg/mL	24 h	HKC (Kidney)	HPLC	Cell extracts	(Downregulation of key enzymes in glycolysis) (Downregulation of major enzymes in this process) (Electron transport chain) (Lysine degradation, tryptophan metabolism, valine, leucine, isoleucine degradation) Amino acid metabolism Energy metabolites Fatty acid metabolism ** Glutamate metabolism Glutathione (GSH) biosynthesis Glycolysis Nucleotide metabolism pathway Nucleotide metabolism * TCA cycle	[84]
	50, 500	100 µg/mL	24 h	Human multi-organ-on-a-chip (DS-MPP) system	UPLC-MS	Huh-7 (Liver)	(Phosphatidic acid, Diacylglycerol, Triglyceride) (Phospholipids) Endocytosis pathway Hepatic lipid metabolism Phospholipase D (PLD) pathway	[52]
	100	10 µg/mL	PND 21	Kunming (Mouse)	LC-MS	Liver Testis	(DHAP, β-D-F6-P, AMP, D-G6-P, TPP) Energy metabolite reduction Hepatic glucose metabolism	[31]
Polystyrene	80	6–125 mg/mL	48 h	L02 (Liver cell)	UPLC-QE Orbitrap MS	Cell extracts	(Associated with TCA cycle, Glutathione metabolism, purine metabolism) Mitochondrial metabolic pathways * (Urea cycle, electron transport chain) NAD^+^, NADH metabolism	[26]
		6–0.25 mg/mL		BEAS-2B (Lung cell)			(Associated with the TCA cycle, Glutathione (GSH, GSSG) Alanine, aspartate, and glutamate metabolism Arginine biosynthesis metabolism, purine metabolism) Mitochondrial metabolic pathways * (Urea cycle, Electron transport chain)	
	20	0.1–10 mg/kg	16 w	Balb/c (Mouse)	LC-MS	Colon	(Fatty acid metabolism and NF-κB signaling)	[19]
				LPS-treated Caco-2		Cell extracts	Fatty acid metabolism ** Glycometabolism Lipid metabolism Lipid peroxidation	
	50	80 µg/mL	24 h	Caco-2 (Colon)	HILIC–LC–MS	Cell extracts	Arachidonic acid metabolism * Drug metabolism by cytochrome P450 Retinol metabolism Steroid hormone biosynthesis *	[54]
		0.1 µg/mL	6 w				Steroid hormone biosynthesis * Terpenoid backbone biosynthesis	
Polystyrene	80	200–1000 µg/L	4 w	C57 BL/6 J (Mouse)	LC-MS	Testis	TCA cycle	[25]
	0.05–0.1	20 mg/kg/day	PND 7~10	Sprague Dawley pup (Rat)	LC–MS	plasma	(glycerol phosphocholines, phosphosphingolipids) Amino acid biosynthesis Aminoacyl-tRNA biosynthesis Sphingolipid metabolism	[85]
Polyethylene terephthalate	200 ± 50	200 mg/kg	14 d	KM (Mouse)	UPLC-MS	Stool	(Arachidonic acid, anserine, and histamine) Lipid metabolites	[30]
	700 ± 300 ^1^						(Sphinganeine)	
	56	0.5 mg/day	28 d	Mus musculus (Mouse)	GC-MS	Stool	Cysteine and methionine biosynthesis D-galactonate degradation Glyoxylate degradation L-alanine biosynthesis L-arginine metabolism Lipid A biosynthesis pathway Sulfate metabolism	[30]
Polycarbonate	30	20, 40 µg/mL	48 h	Second generation UHHS (Liver)	LC-MS	Cell extracts	(Albumin gene downregulation) (CYP2C9)	[51]

^1^ Size definitions of nanoplastics vary across studies. While this review prioritizes 1–1000 nm, some cited papers classify particles slightly exceeding 1 μm as ‘nanoplastics’; we retain the original authors’ terminology here to reflect the source data.

## 3. Metabolomics of Nanoplastic Toxicity in Aquatic Organisms

Pollution by micro- and nanoplastics in aquatic environments has become increasingly evident, and fish are particularly vulnerable due to their anatomical and physiological characteristics [10,86]. While terrestrial organisms are primarily exposed to nanoplastics through food intake, aquatic organisms are additionally exposed via the respiratory system, which makes them a constant threat, and the bioaccumulation is shown to be concentration-dependent [87]. Moreover, the smaller particle size of nanoplastics compared with microplastics enables easier penetration across anatomical barriers and accumulation within internal organs [10]. These factors suggest that aquatic organisms may experience additional or distinct toxicity mechanisms relative to terrestrial species.

From a metabolomic perspective, most aquatic studies have been conducted using integrated multi-omics approaches that combine metabolomic and transcriptomic analyses. Since the changes in metabolites or metabolomic pathways are not binary, but can have different significance, out of the numerous changes reported, this section focuses on representative metabolomic alterations and key pathway-level disruptions commonly identified across aquatic species. An overview of the metabolic pathways and organ-specific responses is shown in Figure 2, providing a visual summary of nanoplastic-induced metabolic reprogramming in aquatic organisms. Detailed datasets and species-specific responses are available in the cited studies and are summarized in Table 2. Most of the metabolic studies covered in this section used a UHPLC-MS or UPLC-MS-based method, and only applications of other analysis techniques will be mentioned.

### 3.1. NPs Toxicity in Aquatic Fish

In the aquatic environment, the first organs of fish that are exposed to nanoplastics are the gills, epidermis, and intestine (considering oral intake). Absorption of nanoplastics is possible in all three; however, most studies report that the intestine and gills are the primary uptake pathways, with the intestine being more susceptible to nanoplastic absorption [88]. After uptake, nanoplastics spread via diffusion to nearby tissues and organs (mostly observed in larvae) or through the circulatory system [10]. Nanoplastics are then accumulated in various organs such as the liver and brain [88]. It is worth mentioning that nanoplastics are small enough to penetrate the BBB and can therefore cause neurological abnormalities. Upon bioaccumulation, nanoplastics cause toxicity that mainly derives from ROS generation, and based on the type of organ, the significant effect can vary.

#### 3.1.1. Gills, Mucus, and Intestine

As demonstrated above, the gills, mucus (epidermis), and intestine are the first organs through which nanoplastics are absorbed by the fish internally. Gills play a crucial role in the respiratory function of fish, and disturbances can lead to oxygen deficiency, resulting in hypoxia stress, which can ultimately cause disturbances in growth, development, and behavior [89]. During the process of respiration, water flows through fish gill filaments and is easily exposed to nanoplastics, which attach to gill surfaces or penetrate the vascular vessel and cell membrane. Due to their smaller size, nanoplastics penetrate more than microplastics, resulting in less direct inflammatory responses in the gills; however, they may be more susceptible to internal accumulation and cause internal damage, such as in the liver or brain [90]. Upon exposure of different sizes of PS-NPs and samples taken from the gills in *O. niloticus* [90], metabolomic analysis showed the disrupted pathways of retrograde endocannabinoid signaling, linoleic acid metabolism, glycerophospholipid metabolism, arachidonic acid metabolism, and arginine and proline metabolism compared to the control group, and when combined with the results of transcriptomics, the arginine and proline metabolism pathway was commonly altered. However, interestingly, although all exposure groups showed reduced adenosine diphosphate (ADP) levels, indicating dysfunction in ATP production and energy supply to the fish’s respiration, the micro-sized plastic showed a greater reduction. The GSH/GSSG (glutathione/oxidized glutathione) ratio, an important indicator of cellular redox balance and oxidative stress, was also reduced by microplastic exposure, but not by nanoplastic exposure. This illustrates that the level of respiratory dysfunction caused in gills is affected by the size (how long the particle stays on the surface of the gills) of the plastic, and that nanoplastic causes relatively less damage due to the ease of elimination (penetration) through the circulatory system. But this also means that internal organs are more susceptible to nanoplastic-derived damage, and permeability should be a concern when investigating nanoplastic toxicity in epidermic cells and tissues.

The skin serves as the primary anatomical barrier, and the secretion of mucus plays a crucial role in the innate immune system. The piscine epidermal mucus is full of active metabolites that contribute to the immune system and the regulation of various metabolisms. Under environmental stress, the composition of the mucus can change as a response. Therefore, the mucus can serve as a sample that reflects the current status of the fish. Also, the mucus can be sampled in a non-invasive manner, while most other samples lead to the death of the specimen, which can have multiple advantages in terms of repeated sampling on the same specimen and is a more cost-effective method considering aquaculture [91]. In a study of polypropylene nanoplastic (PP-NPs) exposure in Nile tilapia, sampling the epidermis mucus and analyzing by OFT-MS/EESI-MS [92], metabonomic analysis showed alterations in the pathways of phenylalanine, tyrosine, and tryptophan biosynthesis, lysine degradation, valine, leucine, and isoleucine biosynthesis, and phenylalanine metabolism. In individual metabolites, histamine, 4-Hydroxynonenal, and caprylic acid were altered. These pathways and metabolites are related to the immunological function of the fish, and considering the role of the mucus, these alterations can impact the overall welfare of the fish and can lead to secondary adverse effects [92].

Considering that the intestine (gut) is an organ that mainly focuses on material (nutrient) intake, its absorption of nanoplastics is the most frequent out of the mentioned organs above, and the toxic effects may be the most significant, because it serves as the entrance of raw material for metabolism [93]. And although most toxicity studies mainly expose waterborne nanoplastics, trophic bioaccumulation is evident, and uptake through dietary sources is also a major pathway, further strengthening the importance [87,94]. On top of the toxicity mechanisms derived from ROS, an increase in inflammation and permeability caused by nanoplastic in the intestine may disturb nutrient absorption and thereby alter the internal metabolism [95]. It is also shown that the gut microbiota is affected by nanoplastics and can lead to disrupted metabolism [96,97]. In contrast to the effects of microplastics on the gills, ROS-mediated stress indicators such as ROS, GSH, superoxide dismutase (SOD), and catalase (CAT) seem to be more intense in comparison in the intestine [98]. This suggests that although they are both epidermic tissues, the effects of nanoplastics differ due to the difference in their absorption, where the intestine targets bigger particles and thus is more susceptible. In a study of polystyrene nanoplastic exposure to *Gobiocypris rarus* sampled in the gut and analyzed by LC-MS [98], metabolomic analysis showed that the pathways of arachidonic acid metabolism, steroid biosynthesis, steroid hormone biosynthesis, tyrosine metabolism, and arginine and proline metabolism were disturbed. Arachidonic acid is a proinflammatory fatty acid and is crucial in maintaining the inflammatory response [99]. This pathway is also important due to the fact that it regulates redox status [100]. Disturbed steroid and steroid hormone biosynthesis can affect the alleviation of inflammation and disturb the modulation of the immune system. Proline has been shown to be an important metabolite related to oxidative stress [101]. In another study of polyvinyl chloride nanoplastics exposed to *Danio rerio* sampled in the gut and analyzed by GC-MS [102], metabolomic analysis showed altered pathways of unsaturated fatty acid biosynthesis, fatty acid elongation and degradation, arginine biosynthesis, starch and sucrose metabolism, the TCA cycle, and pyruvate metabolism. This indicates that nanoplastics have disturbed the energy metabolism and lipid metabolism (glycolipid metabolism).

#### 3.1.2. Liver

After passing the first anatomical barrier, nanoplastics can spread through the circulatory system and accumulate in multiple organs [10]. The liver is a vital organ in various metabolic processes, especially in lipid metabolism and the detoxification process [103,104]. Due to its central role in metabolism, the liver has the second-highest bioaccumulation compared to other organs, where the intestine is first. But considering that the liver is not an epithelial organ, you can estimate that once nanoplastics enter the body, a large portion ends up accumulating in the liver [88]. There were relatively many studies of nanoplastic toxicity in metabolomics that sampled the liver, and this might reflect the importance of the organ. In studies that exposed PS-NPs to *D. rerio* [105,106], metabolic pathways of lipid metabolism, arachidonic acid metabolism, glycerophospholipid metabolism, steroid hormone biosynthesis, purine metabolism, arginine metabolism, phenylalanine, tyrosine, and tryptophan metabolism, glutamine and glutamate metabolism, and ketone metabolism were altered. When looking at individual lipid metabolites, ceramide, sphingomyelin, lysophosphatidylcholine, zymosterol, and triglyceride were altered. In a study of exposure to *Sebastes schlegelii* [107], the metabolic pathways of sphingolipid metabolism, ABC transporters, protein digestion and absorption, TCA cycle, ubiquinone metabolism, and cell apoptosis-related pathways (ferroptosis and FoxO signaling) were altered. In a study on *Hypophthalmichthys molitrix* [108], the metabolic pathways of ABC transporters, glycerophospholipid metabolism, aminoacyl-tRNA biosynthesis, and arginine biosynthesis were altered. In a study on *Sciaenops ocellatus* [109], the metabolic pathways of glycerophospholipid metabolism, choline metabolism in cancer, sphingolipid signaling, retrograde endocannabinoid signaling, pyrimidine metabolism, purine metabolism, taurine and hypotaurine metabolism, and ABC transporters were altered. In a study of PP-NPs exposure to Nile tilapia analyzed by OFT-MS/EESI-MS [110], the metabolic pathways of glycerophospholipid metabolism, arginine and proline metabolism, and aminoacyl-tRNA biosynthesis were altered. Lipid metabolism (specifically sphingolipid and glycerophospholipid), ABC transporters, arginine and purine metabolism, and aminoacyl-tRNA biosynthesis pathways were commonly mentioned and discussed. Sphingolipids are lipids that contain a backbone of sphingoid bases such as ceramides and sphingomyelin [111]. Each class of Sphingolipid has a different role, where ceramide plays an important role in cell growth, proliferation, and apoptosis, while sphingomyelin interacts with cholesterol and promotes lipid raft formation and participates in signal transduction [112,113]. The ABC transporter family is important in actively transporting various substances, including lipids, drugs, and toxins [114]. Reduced efficiency due to the disturbance may lead to lower rates of nanoplastic efflux and ultimately cause intracellular accumulation. Arginine is a multifunctional amino acid. It is a precursor of ornithine, urea, creatine, and other substances and has been proposed as an indicator for liver injury [115]. It is also important in cell proliferation and can scavenge free radicals [116,117]; thus, the alterations may be a result of nanoplastic-induced stress. Purines are nitrogen-containing heterocycles that have an important role in being the building components of DNA, RNA, and coenzymes. They can also function as a direct neurotransmitter, and specific types, such as xanthine, can produce ROS in their metabolism [118]. Aminoacyl-tRNA biosynthesis is the direct acylation of tRNA and amino acids, which is related to polypeptide and protein synthesis [119].

#### 3.1.3. Brain

The brain is a complex organ that closely functions and physically communicates with the other organ systems. Because of its significance, it is protected by a highly impermeable innate barrier, the BBB. The BBB effectively excludes many unnecessary soluble plasma contents and acts as an active barrier to many toxic xenobiotics [120]. However, nanoplastics can pass the BBB and accumulate in the brain and central nervous system (CNS), resulting in neuron damage, CNS inflammation, and neurodegenerative diseases [74,121,122]. The brain is closely related to the digestive system through the gut–brain axis [123]. This axis is a complex bidirectional network between the gut-intestine tract and the CNS. It incorporates microbial, neuronal, immune, and hormonal pathways that have a major impact on the welfare of the specimen. Neurotransmitters, neuropeptides, and immune signals produced in the gut may influence the activity of the brain, and metabolism can change if the microbiota alters [124]. In a study of polystyrene nanoplastic exposure to *D. rerio* sampled in the brain and analyzed by UHPLC-MRM-MS/MS and LC-MS/MS [96,125], the metabolic pathways of tryptophan metabolism, phenylalanine metabolism, nitrogen metabolism, arginine and proline metabolism, alanine metabolism, aminoacyl-tRNA biosynthesis, glycine, serine and threonine metabolism, phenylalanine, tyrosine and tryptophan biosynthesis, aspartate and glutamate metabolism, and methane metabolism were altered. In terms of individual metabolites, neurotransmitter function-related metabolites such as 5-hydroxytryptophan, 5-hydroxyindoleacetic acid, histamine, homovanillic acid, L-asparagine, L-cysteine, L-glutamic acid, L-phenylalanine, L-tryptophan, norepinephrine, and serotonin were altered. Tryptophan, phenylalanine, and glutamic acid (glutamate) are the commonly mentioned and discussed metabolites. Tryptophan is an essential amino acid in development and is supplied through diet. Circulating tryptophan can activate the CNS astrocytes but also suppresses CNS inflammation and can be used as a precursor for neurotransmitters such as serotonin [126]. The microbiota can metabolize tryptophan into aryl-hydrocarbon receptor ligands, which can signal an immune response [127]. Phenylalanine is a precursor of tyrosine, which is the precursor for neurotransmitters such as dopamine, norepinephrine, etc. Glutamate is the anionic form of glutamic acid in the body and acts as a bridge connecting glucose and amino acid metabolism [125]. It can also restore depleted glutathione under oxidative stress [128].

#### 3.1.4. Early Development Stage

Fish in early larval stages grow rapidly; are mostly transparent, which makes them easy to observe; are more susceptible to xenobiotics; and can give insights into the effects on development. For these reasons, many researchers use fish larvae as an experimental model. But despite the above, considering that most habitats are to some degree polluted by micro- or nanoplastic, the chronic exposure effects, including vertical transmission, effects on embryos and early development, should be investigated. In a study of PS-NP exposure to Oreochromis mossambicus larvae [129], the metabolic pathways of fatty acid synthesis, metabolism of alpha-linolenic acid, arachidonic acid metabolism, biosynthesis of unsaturated fatty acids, pentose phosphate metabolism, TCA and glycolysis/gluconeogenesis, and amino acid metabolism (L-pipecolic acid, phenylacetyl glycine, keto-isocaproic acid) were altered. In a study of PS-NPs exposure to D. rerio larvae analyzed by GC-TOF/MS [130], the metabolic pathways of steroid hormone biosynthesis, purine metabolism, alanine, aspartate, glutamate metabolism, citrate cycle, and Galactose metabolism were altered. In a study of PET-NPs exposure to D. rerio larvae analyzed by NMR [131], the metabolic pathways of lipid metabolism, membrane lipid, phenylalanine, tyrosine, and tryptophan metabolism, glutathione metabolism, and the TCA cycle were altered. Commonly, fatty acid metabolism and the TCA cycle were mentioned and discussed. Fatty acids are a broad term for various lipids and contribute to many biological functions, such as energy metabolism and cellular structure. Isoleucine, valine, and leucine are branched-chain amino acids that are known to promote fatty acid metabolism and prevent fat accumulation. Therefore, a disturbance in those levels may affect the metabolism of fatty acids.

### 3.2. NPs Toxicity in Aquatic Invertebrates

Nanoplastic uptake and bioaccumulation through trophic pathways are evident, and since most aquatic invertebrates are lower in the food chain, it is crucial to understand the effects on them [94]. Despite their ecological importance, relatively few studies have applied metabolomic approaches to assess nanoplastic toxicity in aquatic invertebrates, and most available work involves PS-NPs. Although several metabolomics studies exist for microplastic exposures (e.g., 20 µm MPs) [132], research using true nanoscale particles and diverse polymer types remains limited. Future studies should incorporate nano-sized plastics and a wider range of environmentally relevant polymers to more accurately represent real-world pollution scenarios.

Many aquatic invertebrates, such as clams, krill, and zooplankton, are filter feeders. This inevitably exposes them to waterborne nanoplastics more frequently than non-filter feeders and even more compared to surface breathers [133]. Although it has also been reported that several filter-feeding copepods and bivalves are known to distinguish non-edible particles and can selectively feed, other species like *Daphnia magna* can be more susceptible due to their feeding characteristics [134]. In a study of PS-NPs exposure to *D. magna* larvae analyzed by NMR [135], the following metabolites were altered: lysine, phenylalanine, tyrosine, and isopropanol. Lysine is discussed to be related to molting, and isopropanol is produced by microorganisms, which may indicate that the bacterial conversion is affected [136]. In a study of marine rotifers (*Brachionus plicatilis*) exposed to PS-NP [137], the pathways of alanine, aspartate, and glutamate metabolism, TCA, purine metabolism, and pyrimidine metabolism were altered. In a study of triploid Fujian oyster (*Crassostrea angulata)* exposed to PS-NP [138], the pathways of arachidonic acid metabolism, arginine and proline metabolism, glutathione metabolism, lysine degradation, phenylalanine, tyrosine, and tryptophan biosynthesis, fructose and mannose metabolism, ubiquinone and other terpenoid-quinone biosynthesis, thiamine metabolism, and purine metabolism were altered. In a study of *Paphia undulata* (marine clam) exposed to PS-NP analyzed by GC-TOF/MS [139], the phenylalanine, tyrosine, and tryptophan biosynthesis, galactose metabolism, starch/sucrose metabolism were altered. *Cherax quadricarinatus* is a largely aquacultured crayfish that is omnivorous and a filter feeder [140,141]. In a study of PS-NPs exposure to *C. quadricarinatus* larvae [142], the pathways of cholesterol metabolism, ovarian steroidogenesis, peroxisome proliferator-activated receptor (PPAR) signaling, insect hormone biosynthesis, bile secretion, and phenylalanine metabolism were altered. PPARs are a group of nuclear receptor proteins that function as transcription factors and are important in regulating different metabolisms [143].

**Table 2 ijms-27-00050-t002:** Metabolomic studies of different nanoplastic toxicity in aquatic organisms (in vivo). Each study reported various metabolic pathways, but only the main discussed pathways are represented in ‘Key pathways’. Asterisks mean that a pathway was commonly reported in multiple studies (* 2 studies, ** more than 3 studies). Brackets in ‘key pathways’ mean either that a metabolite was reported but was not discussed in a specific pathway or that multiple metabolites were described in a vague group.

Nano Particles	Size (nm)	Concentration	Exposure Time	Species	Method	Sample	Key Pathways	Reference
Polystyrene	100 ± 5	20 mg/L	7 d	*O. mossambicus* (Tilapia)	UPLC-Q -TOF-MS	Whole body (Larvae)	Alpha-Linolenic acid metabolism Arachidonic acid metabolism ** Pentose phosphate metabolism Synthesis of fatty acids TCA glycolysis/gluconeogenesis ** Unsaturated fatty acids biosynthesis *	[129]
	50–100	0.1–10 mg/L	116 hpf	*D. rerio* (Zebra fish)	GC-TOF/MS	Whole body (Larvae)	Alanine, aspartate, and glutamate metabolism ** Citrate cycle ** Galactose metabolism * Glycerolipid metabolism * Purine metabolism ** Steroid hormone biosynthesis *	[130]
	42–44	1–100 µg/L	30 d	*D. rerio*	UHPLC-MRM-MS/MS	Brain	(Neurotransmitter-related metabolites) Phenylalanine metabolism ** Tryptophan metabolism *	[96]
	51 ± 3, 52 ± 5	30–50 mg/L	120 hpf	*D. rerio*	UHPLC-MS/MS	Whole body (Larvae)	(Neurotransmitter-related metabolites) Glutathione metabolism ** Polyamine metabolism Tryptophan-kynurenine pathway *	[144]
Polystyrene	143.36 ± 5.11	230 µg/L	15 d	*S. schlegelii* (Jacopever)	UHPLC-MS/MS	Liver	ABC Transporters ** Cell apoptosis pathways Protein digestion and absorption Sphingolipid metabolism * TCA cycle ** Ubiquinone metabolism *	[107]
	80	15–150 µg/L	21 d	*D. rerio*	UHPLC-MS/MS	Liver	Lipid metabolism (Ceramide, sphingomyelin, lysophosphatidylcholine, zymosterol, TG)	[105]
	50.85 ± 8.74, 46.37 ± 0.49	3.2–320 µg/L	37 d	*D. magna*	NMR	Whole body	(Lysine) Catecholamine synthesis	[135]
	80	10–1000 µg/L	96 h	*H. molitrix* (Carp)	UHPLC-MS/MS	Liver	ABC Transporters ** Aminoacyl-tRNA biosynthesis ** Arginine biosynthesis * Glycerophospholipid metabolism **	[108]
	100	100–1000 µg/L	28 d	*C. quadricarinatus* (Cray fish)	UPLC-MS/MS	Hepatopancreas	Bile secretion Cholesterol metabolism Insect hormone biosynthesis Ovarian steroidogenesis Phenylalanine metabolism ** PPAR signaling	[142]
Polystyrene	109.1 ± 1.6	1–10 mg/L	14 d	*G. rarus* (Minnow)	LC-MS	Intestine	Arachidonic acid metabolism ** Arginine and Proline metabolism ** Steroid biosynthesis Steroid hormone biosynthesis * Tyrosine metabolism	[98]
	29.15 ± 8.07	5 mg/L	7 d	*S. ocellatu* (Red drum)	UHPLC-MS	Liver	ABC transporters ** Choline metabolism in cancer Glycerophospholipid metabolism ** Purine metabolism ** Pyrimidine metabolism * Retrograde endocannabinoid signaling * Sphingolipid signaling pathway * Taurine and hypotaurine metabolism	[109]
	78.1 ± 0.4, 81.2 ± 0.9	100 µg/L	28 d	*O. niloticus* (Tilapia)	UHPLC-MS/MS	Gill	(ADP) Arachidonic acid metabolism ** Arginine and Proline metabolism ** Glycerophospholipid metabolism ** Linoleic acid metabolism Retrograde endocannabinoid signaling *	[90]
	50 ± 10	0.1–10 mg/L	28 d	*D. rerio*	UHPLC/Q-TOF MS	Liver	Arachidonic acid metabolism ** Arginine metabolism ** Glutamine and glutamate metabolism Glycerophospholipid metabolism ** Ketone body synthesis and degradation Phenylalanine, tyrosine, and tryptophan metabolism ** Purine metabolism ** Steroid hormone synthesis **	[106]
Polystyrene	44	10 µg/L	120 d	*D. rerio*	LC-MS/MS	Brain	Alanine, aspartate, and glutamate metabolism ** Aminoacyl-tRNA biosynthesis ** Arginine and Proline metabolism ** Glycine, serine, and threonine metabolism Methane metabolism Phenylalanine, tyrosine, and tryptophan biosynthesis **	[125]
	100	10–104 particles/L	14 d	*C. angulata* (Triploid oyster)	UHPLC-MS/MS	Hepatopancreas	Arachidonic acid metabolism ** Arginine and proline metabolism ** Fructose and mannose metabolism Glutathione metabolism ** Lysine degradation * Phenylalanine, tyrosine and tryptophan biosynthesis ** Purine metabolism ** Thiamine metabolism Ubiquinone and other terpenoid -quinone biosynthesis *	[138]
Polystyrene	80	500 µg/L	4 d	*P. undulata* (Marine clam)	GC-TOF/MS	Digestive glands	Phenylalanine, tyrosine and tryptophan biosynthesis Galactose metabolism * Starch and sucrose metabolism *	[138]
	70–500	200 µg/L	48 h	*B. plicatilis* (Rotifer)	UHPLC-MS/MS	Whole body	Alanine, aspartate and glutamate metabolism ** TCA ** Purine metabolism ** Pyrimidine metabolism *	[137]
Polypropylene	100	1–100 mg/L	21 d	Nile tilapia	OFT-MS/EESI-MS	Mucus	(4-Hydroxynonenal) (Caprylic acid) (Histamine) Lysine degradation * Phenylalanine metabolism ** Phenylalanine, tyrosine, and tryptophan biosynthesis ** Valine, leucine, and isoleucine biosynthesis *	[92]
	100	1–100 mg/L	21 d	Nile tilapia	OFT-MS/EESI-MS	Liver	Aminoacyl-tRNA biosynthesis ** Arginine and Proline metabolism ** Glycerophospholipid metabolism **	[110]
Polyethylene terephthalate	70 ± 5	100 mg/L	24 d	*D. rerio*	NMR	Whole body	(Trimethylamine N-oxide) (Membrane lipid) Glutathione metabolism ** Phenylalanine, tyrosine, and tryptophan biosynthesis ** TCA ** Valine, leucine, and isoleucine biosynthesis *	[131]
Polyvinyl chloride	200	50 µg/L	18 d	*D. rerio*	GC-MS	Gut	Arginine biosynthesis * Fatty acid elongation and degradation Pyruvate metabolism Starch and sucrose metabolism * TCA ** Unsaturated fatty acids biosynthesis *	[102]

## 4. Comparative Metabolic Signatures Across Mammalians and Aquatic Organisms

Metabolomics has an advantage over other omics methods in that metabolite structures are relatively conserved across species, compared to the nucleic acid and amino acid sequences of genes and proteins [37]. Metabolic pathways also show a substantial degree of interspecies conservation [145]. Despite the anatomical and evolutionary differences between mammalian and aquatic organism models, several metabolic pathways were similarly affected following nanoplastic exposure. Cross-species comparison therefore provides an opportunity to identify shared metabolic vulnerabilities and to contextualize species-specific responses. Across all sampling targets, mammals most frequently exhibited alterations in fatty acid metabolism, nucleotide metabolism, the TCA cycle (including electron transport chain dysfunction), and arachidonic acid metabolism. In aquatic organisms, commonly reported pathways included the TCA cycle, glycerophospholipid metabolism, arachidonic acid metabolism, arginine and proline metabolism, and phenylalanine, tyrosine, and tryptophan metabolism. The TCA cycle and arachidonic acid metabolism represented the major intersections between the two model types. Table 3 summarizes organ-specific metabolic disruptions and highlights the common pathways identified across mammals and aquatic organisms.

### 4.1. Common Metabolic Pathways by Organs

Due to the fundamental difference between mammals and aquatic organisms, there were some variations in sampling targets. The liver, intestine (gut), and brain were common to both model types, whereas lung–gill and fetus–larvae comparisons reflect analogous biological roles rather than identical organs. The mucus is an abundant secretion in both mammals and aquatic organisms and possibly a common target, but when limiting to the epidermal mucus used in fish studies, the internal mucus of mammalians should be considered distinct, and no studies of nanoplastic toxicity using metabolomics could be found. In the liver, glycerophospholipid metabolism was a consistently altered pathway in both mammals and aquatic organisms. In the intestine, arachidonic acid metabolism and steroid-related metabolic pathways showed common disturbances. In the brain, glutamate metabolism and tyrosine metabolism were generally altered. Although no identical key pathways were reported between lungs and gills, this absence does not indicate biological irrelevance. Because only the major pathways highlighted in each study were compared here, additional minor or less-emphasized metabolic changes may still show overlap. Finally, both the mammalian fetus and aquatic larvae commonly exhibited alterations in the TCA cycle, highlighting energy metabolism as a shared vulnerability during early developmental stages.

### 4.2. Linkage Between Nanoplastic Toxicity Mechanisms and Key Pathway Alterations

As mentioned earlier, the metabolic pathways in which changes commonly occur in aquatic organisms and mammals were glycerophospholipid metabolism, arachidonic acid metabolism, and the TCA cycle. Disturbance in these pathways leads to a range of toxic effects and pathological changes. Figure 3 summarizes the mechanistic relationships among these pathways.

Glycerophospholipids are essential structural lipids of cell membranes and play key roles in maintaining membrane stability and cellular signaling [146]. They are largely classified into phosphatidylcholine (PC), phosphatidylethanolamine (PE), phosphatidylserine (PS), phosphatidylinositol (PI), phosphatidic acid (PA), phosphatidylglycol (PG), and cardiolipin (CL). The glycerophospholipid metabolic pathway regulates the biosynthesis, degradation, and remodeling of key phospholipids (PC, PE, PS, PI, etc.). Disturbance of glycerophospholipid metabolism due to nanoplastic exposure impairs membrane stability, defense functions, and membrane fluidity, and alters the geometric properties of the lipid bilayer. These changes affect receptor activity, ion channel function, and intracellular signaling [147,148,149]. The significant increase in saturated PC species—such as 1-palmitoyl-sn-glycero-3-phosphocholine (PC 16:0), 1-stearoyl-sn-glycero-3-phosphocholine (PC 18:0), and the remodeling marker LysoPC 18:0—together with the accumulation of saturated fatty acids (e.g., palmitic acid; C16:0), strengthens membrane packing and increases bilayer rigidity. Conversely, the decrease in PUFA-containing phospholipids, including 1-oleoyl-sn-glycero-3-phosphocholine (PC 18:1), myristoleic acid (C14:1), and long-chain PUFAs such as arachidonic acid (AA), linoleic acid (LA), and docosahexaenoic acid (DHA), further restricts membrane fluidity. This shift toward a more rigid membrane state enhances cellular stress and promotes activation of cell death pathways. In mitochondria, excessive membrane rigidity disrupts oxidative phosphorylation and ATP production, ultimately impairing neuronal synaptic function [80,150].

Membrane rigidity is also regulated by the PE/PC ratio [151]. PE is a cone-shaped lipid that forms hexagonal (HII) structures in aqueous environments, and this property is essential for maintaining membrane dynamics and fluidity [152]. In addition, PE is present in the inner leaflet, and the composition of the inner leaflet can affect the fluidity of the outer leaflet [153,154]. PC is a cylindrical lipid and the most abundant double-layer-forming lipid in most cell membranes [155]. Due to these properties, a decreased PC/PE ratio (relative accumulation of PE) creates negative curvature stress, leading to lipid packing defects. This structural instability compromises the barrier function, resulting in increased membrane permeability and the leakage of intracellular contents [155,156]. Conversely, an increased PC/PE ratio (relative accumulation of PC) leads to excessive membrane stabilization. This reduces the membrane’s ability to undergo necessary curvature changes, thereby impairing essential dynamic processes such as vesicle fusion/fission and mitochondrial respiration [157]. Therefore, the PC/PE ratio is an important regulator of membrane integrity, and its imbalance induces toxicity such as lactate dehydrogenase leakage and inflammatory reactions [151,156,158]. Especially reduced GP unsaturation in neuronal membranes impairs neurotransmitter-related processes, which in turn affects emotional and cognitive functions [159]. Decreased GP unsaturation and reduced DHA-PE/PC and AA-PE/PC levels increase neuronal membrane stiffness, impair synaptic vesicle function, and ultimately suppress neurotransmitter release [160].

In addition, PA deficiency due to glycerophospholipid metabolism disturbance may lead to a decrease in CoA levels, which induces selective inactivation and/or decreased synthesis of enzymes that catalyze major steps in fatty acid oxidation in peroxisomes [161]. Because PA is also required for glutathione (GSH) synthesis, reduced PA levels lead to decreased GSH availability. Thus, a decrease in PA intensity may affect fatty acid oxidation pathways and cause oxidative stress [161].

Arachidonic acid (AA) is an unsaturated fatty acid stored in cell membrane phospholipids and is a physiologically active lipid that is key to the regulation of inflammation, immunity, and nerve function. It is esterified and stored in the form of PC or PE in the cell membrane, and then is cleaved and released from the membrane phospholipid when phospholipase (PLA_2_), which is a phospholipid hydrolase, is activated. The released AA enters the COX, LOX, and CYP450 pathways and is converted into potent inflammatory and immune mediators such as PGE_2_, leucotriene, and thromboxylic acid (TXA_2_). When arachidonic acid metabolic disturbance occurs due to nanoplastic exposure, the expression and secretion of IL-1b, IL-6, and IL-8 are abnormally increased or decreased. In addition, the oxidative stress of cells itself has a dramatic effect on AA metabolism, which alters AA release by PLA2, promotes autoxidation of AA, and induces COX2 overexpression [162]. This shifts the Eicosanoid profile toward pro-inflammatory/pro-tumorigenic mediators and synthesizes sub-products such as prostaglandin to induce local inflammation [162,163]. Consequently, disruption of arachidonic acid metabolism may promote cancer development and progression.

The tricarboxylic acid (TCA) cycle is a central mitochondrial pathway that links carbohydrate, lipid, and amino acid metabolism and provides reducing equivalents for ATP production [164]. It has been reported that abnormalities in these energy metabolism processes, especially the decline in the function of the TCA circuit, are closely related to various neurological diseases [165,166]. Since most of the ATP required by neurons is generated during oxidative phosphorylation through mitochondrial TCA circuits, damage to the TCA circuit has a direct adverse effect on neural function. Nanoplastic exposure causes a decrease in TCA cycle activity, resulting in a decrease in the level of TCA intermediates (M + 2 and M + 4) and mRNA expression of associated key enzymes (Pdha1, Cs, Idh2, Ogdh). This decreases the NADH supplied to the electron transport system (ETC) and eventually results in a decrease in ATP production. In addition, as the reducing power of cells (NADPH/NADH) decreases, ROS accumulation increases, resulting in deterioration of mitochondrial structure and function. This energy depletion and oxidative damage can cause various reproductive and developmental disorders, such as cytoskeletal abnormalities, meiosis failure, and fertilization rate reduction. TCA cycle collapse causes cognitive and memory loss in the hippocampus of mice and histologically leads to nuclear enrichment, cell count reduction, and synapse loss [166]. In the zebrafish model, energy metabolic disorders caused pericardial edema, mandibular developmental abnormalities, decreased motor capacity, hindered egg maturation, and caused early cell death [167]. Collectively, these findings support a unified mechanism in which nanoplastic-induced oxidative stress and mitochondrial impairment drive disruptions in glycerophospholipid metabolism, arachidonic acid signaling, and the TCA cycle.

As highlighted in the preceding sections, nanoplastic exposure induces cellular inflammatory responses and elevates ROS levels, thereby disrupting various metabolic pathways. Specifically, increased ROS acts as a proximal signal that activates MAPK cascades (ERK, JNK, p38 MAPK) and triggers the NF-κB transcriptional program [168,169,170]. MAPK signaling governs critical cellular activities, including gene expression, apoptosis, inflammation, and the regulation of immune responses [168], while directly facilitating NF-κB activation [171]. NF-κB is a transactivator that induces transcription of various target genes in an immediate response to a pathogen or harmful stimulus. NF-κB is also present in unstimulated cells, and when the cells are stimulated, the nuclear migration signal of the NF-κB dimer is exposed, which moves to the nucleus and binds to the DNA [171]. NF-κB regulates the expression of several genes, such as inflammatory cytokines (TNF-α, IL-1β, etc.), chemokines (IL-8 and others), and hematopoietic growth factors [168,171], and exacerbates inflammatory damage by mediating the secretion of cytokines such as tumor necrosis factor-α (TNF-α), interleukin-1β (IL-1β), and interleukin-6 (IL-6).

Crucially, this inflammatory signaling bridges the gap to metabolic dysregulation. For instance, the induction of TNF-α upregulates Phospholipase A2 (PLA2) [172]. PLA2 is the enzyme that hydrolyzes membrane phospholipids at the *sn-2* position to release free arachidonic acid (AA). This liberated AA is subsequently metabolized by COX-2—whose expression is also induced by NF-κB—into potent eicosanoids. This cascade amplifies oxidative stress and inflammation [173], ultimately culminating in distinct forms of cell death, including necroptosis, pyroptosis, and apoptosis [174]. Thus, the signaling pathways activated by oxidative stress serve as the mechanistic drivers of the lipid and energy metabolism disruptions observed in this review.

### 4.3. Translational Applications of Metabolomic Findings in Nanoplastic Toxicology

Metabolomics has valuable translational potential across diverse fields, including medicine, food science, environmental sciences, agriculture, and toxicological research [158]. In human health assessment, metabolomics plays a central role in disease diagnosis, patient stratification, and therapeutic monitoring, and is widely applied in oncology through the identification of cancer-associated metabolic biomarkers [159]. In environmental toxicology, metabolomics enables exposome-level evaluations by linking pollutant-induced metabolic fingerprints to specific classes of contaminants such as heavy metals and pesticides [160,161].

Within the context of nanoplastic toxicity, these principles suggest several potential translational applications. First, recurrent metabolic alterations—such as disruptions in glycerophospholipid metabolism, TCA cycle imbalance, or oxidative stress-related amino acid pathways—may serve as candidate biomarkers of nanoplastic exposure. With further validation, such signatures could be incorporated into targeted metabolite-based screening panels or field-deployable assays for environmental monitoring. Second, dose-dependent metabolic alterations may contribute to establishing biologically relevant thresholds for early cellular distress, thereby informing future efforts in human health risk assessment or the derivation of safe exposure limits. Finally, species-specific metabolic fingerprints could complement existing ecotoxicological indices and provide more sensitive tools for evaluating the ecological impact of environmental nanoplastics. While these applications remain preliminary due to limited datasets and the early stage of nanoplastic–metabolomics research, they highlight the growing potential of metabolomics to bridge mechanistic toxicology with regulatory and environmental monitoring frameworks. In addition, the emerging translational implications of metabolomic signatures—such as their potential use in environmental monitoring and biomarker development—highlight the importance of expanding metabolomic research in future nanoplastic toxicology.

## 5. Conclusions and Perspectives

Nanoplastic pollution represents an emerging threat to environmental and human health, with increasing evidence of metabolic disruption across mammalian and aquatic organisms. The metabolomics-based studies reviewed here consistently demonstrate that nanoplastic exposure perturbs key metabolic domains—including energy metabolism, lipid metabolism, amino acid pathways, and nucleic acid-related processes. Among these, the tricarboxylic acid (TCA) cycle and arachidonic acid metabolism were the most recurrently altered pathways across tissues and species, highlighting shared metabolic vulnerabilities.

Despite these interesting findings, some limits in the current state of research should be considered in future studies. First, most mechanistic evidence is derived from studies using polystyrene nanoplastics (PS-NPs), which may not represent the diversity of environmentally relevant nanoplastic types. Broader investigation across different polymer types is therefore needed. Secondly, although metabolomics already plays a central role in this field, its application remains limited in depth and scope compared with transcriptomics. Future studies should expand metabolomics-based analyses—particularly within multi-omics frameworks—to more comprehensively elucidate the biochemical mechanisms of nanoplastic toxicity. Such integration will help clarify how gene-expression changes translate into metabolic outcomes and toxic phenotypes. Lastly, taxonomic coverage remains narrow. Most studies rely on traditional laboratory models such as mice and zebrafish, whereas numerous ecologically and economically important aquatic species remain unexplored, despite being highly vulnerable to nanoplastic contamination.

Looking forward, future research should extend current metabolomics findings by integrating multi-omics approaches—including transcriptomics, lipidomics, proteomics, and microbiomics—to establish clearer mechanistic links between molecular signaling pathways and downstream metabolic phenotypes. Such integration will help explain how nanoplastic-induced oxidative stress, inflammatory signaling, and mitochondrial dysfunction translate into disruptions in glycerophospholipid, arachidonic acid, and energy metabolism. Additionally, the field would benefit from adopting more realistic exposure models that reflect environmental complexity, including weathered particles, mixed-polymer nanoplastics, and biofilm-coated particles that better represent real-world pollution. Expanding research to a wider taxonomic range, particularly ecologically relevant aquatic invertebrates and commercially important species, will also enhance the ecological relevance of future conclusions.

Moreover, metabolomic signatures altered by nanoplastic exposure have strong translational potential. Identifying robust metabolic biomarkers could support the development of monitoring tools for environmental assessment, early-warning systems for ecological stress, and metabolite-based assays to evaluate human exposure and susceptibility. To advance such applications, methodological standardization—including harmonized sample collection, storage protocols, detection platforms, and quality-control procedures—will be essential to improve reproducibility and allow cross-study comparisons. Continued progress in these areas will enable metabolomics to evolve from descriptive profiling toward actionable insight, supporting evidence-based strategies for ecological risk evaluation and human health protection in the face of growing nanoplastic contamination.

In conclusion, nanoplastics pose a growing threat to both environmental and human health. Advancing metabolomics-based research, in combination with other omics technologies, will be crucial for decoding the complex biochemical mechanisms of nanoplastic toxicity and for guiding effective strategies in ecological risk assessment and pollution management.

## Figures and Tables

**Figure 1 ijms-27-00050-f001:**
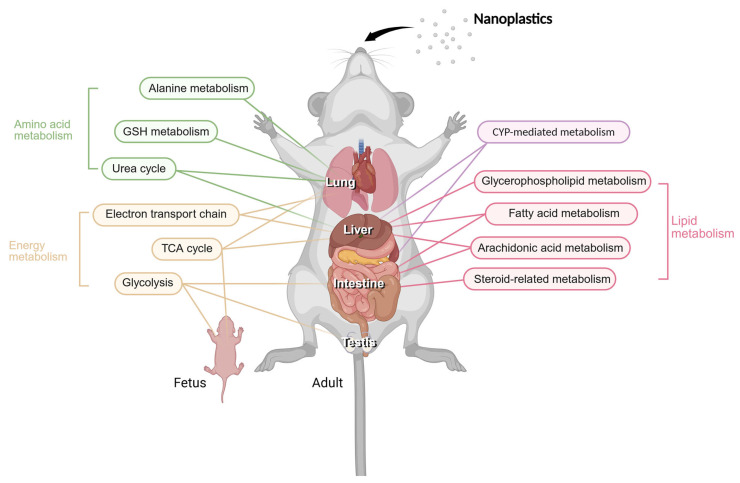
Summary of the mainly discussed metabolic pathway alterations in different mammalian organs following nanoplastic exposure. Pathways are color-coded as follows: pink for lipid metabolism, blue for nucleotide metabolism, yellow for energy metabolism, green for amino acid metabolism, and purple for other pathways.

**Figure 2 ijms-27-00050-f002:**
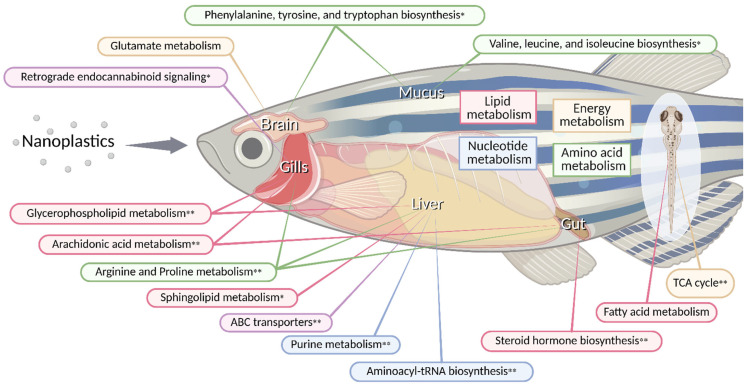
Summary of the mainly discussed metabolic pathway alterations in different fish organs following nanoplastic exposure. Asterisks indicate pathways commonly reported across multiple studies (* in 2 studies, ** in more than 2 studies). Pathways are color-coded as follows: pink for lipid metabolism, blue for nucleotide metabolism, yellow for energy metabolism, green for amino acid metabolism, and purple for other pathways.

**Figure 3 ijms-27-00050-f003:**
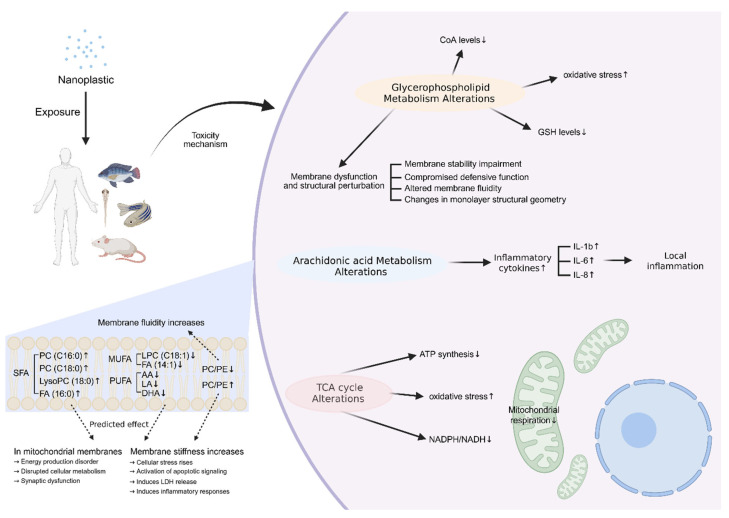
Summary of metabolomic alterations and toxicological responses following nanoplastic exposure. Solid arrows indicate causal relationships supported by experimental evidence, whereas dashed arrows represent predicted changes inferred from integrated findings of previous studies. Directional arrows (↑, ↓) denote increases or decreases in the indicated parameters as described in the main text.

**Table 3 ijms-27-00050-t003:** Comparative metabolomic findings describing key pathway alterations induced by nanoplastic exposure in mammals and aquatic organisms. Common metabolic pathways between matching organs (sample targets) are shown in bold.

Organ (Sample)	Model	Key Pathways
Liver	Mammals	Arachidonic acid metabolism CYP-mediated metabolism Electron transport chain Fatty acid metabolism **Glycerophospholipid metabolism** TCA cycle Urea cycle
	Aquatic organisms	ABC Transporters Aminoacyl-tRNA biosynthesis Arginine and Proline metabolism **Glycerophospholipid metabolism** Purine metabolism Sphingolipid metabolism
Intestine (Gut)	Mammals	**Arachidonic acid metabolism** CYP-mediated metabolism Fatty acid metabolism **Steroid-related metabolism** TCA cycle
	Aquatic organisms	**Arachidonic acid metabolism** Arginine and Proline metabolism **Steroid hormone biosynthesis**
Brain	Mammals	Cysteine metabolism **Glutamate metabolism** Glycerolipid metabolism Glycine, serine, and threonine metabolism Leucine, isoleucine, valine metabolism Sphingolipid metabolism **Tyrosine metabolism**
	Aquatic organisms	**Glutamate metabolism** **Phenylalanine, tyrosine, and tryptophan metabolism**
Lungs	Mammals	Alanine metabolism Electron transport chain Glutathione metabolism TCA cycle Urea cycle
Gills	Aquatic organisms	Arachidonic acid metabolism Arginine and Proline metabolism Glycerophospholipid metabolism Retrograde endocannabinoid signaling
Fetus	Mammals	**TCA cycle** Glycolysis
Larvae	Aquatic organisms	**TCA cycle** Fatty acid metabolism

## Data Availability

No new data were created or analyzed in this study.

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
