# Peer review of "Comparative Metabolomic Approaches to Nanoplastic Toxicity in Mammalian and Aquatic Systems"

_ijms, 2025, doi:10.3390/ijms27010050_

Round 1
Reviewer 1 Report
Comments and Suggestions for Authors
Some comments and suggestions major revision are listed below.
1. Resolve inconsistencies in nanoplastic size definitions and table annotations to enhance precision. The manuscript initially defines NPs as 1 nm–1 μm (Line 41–42), aligning with metabolomics studies. However, in Table 2 (e.g., PS-NPs with size 50–100 nm, PET-NPs with 700±300 nm), some "nanoplastics" exceed 1 μm (e.g., 700±300 nm overlaps with microplastics). This inconsistency confuses readers and undermines the review’s precision. Recommendation: - Add a footnote to Table 2 clarifying that particles exceeding 1 μm are included due to their classification in cited studies, while reaffirming the manuscript’s 1 nm–1 μm definition. - In the Introduction (Section 1), explicitly acknowledge that size classifications vary across studies and explain why the 1 nm–1 μm range is prioritized (e.g., consistency with metabolomic research).
2. Strengthen cross-species comparative analysis by adding a dedicated subsection and visual summary: The manuscript separately discusses mammalian and aquatic systems but lacks a dedicated section comparing shared vs. distinct metabolic disruption patterns. For example, both systems exhibit lipid metabolism disturbance, but aquatic organisms (e.g., fish gills) show unique mucus metabolism alterations that are not contrasted with mammalian barriers (e.g., intestinal epithelium). Recommendation: - Add a new subsection (e.g., "3.4 Comparative Metabolic Signatures Across Mammalian and Aquatic Systems") to synthesize: - Shared pathways (e.g., arachidonic acid metabolism, TCA cycle impairment). - Species-specific differences (e.g., mucus metabolism in aquatic organisms vs. gut-liver axis disruption in mammals). - Use a concise table (e.g., Table 3) to list key perturbed pathways, organs, and species, enhancing visual comparison.
3. Ambiguous Description of Mechanistic Links: The manuscript repeatedly mentions that NPs induce oxidative stress and mitochondrial dysfunction, but the mechanistic link between these processes and specific metabolic changes is unclear. Recommendation: - For key pathways (e.g., TCA cycle, lipid peroxidation), add a brief mechanistic explanation using cited studies. Use a simple schematic (e.g., Figure 3) to illustrate the cascade: NP exposure → ROS/mROS → mitochondrial dysfunction → metabolic pathway disruption.
4. Typographical Errors: Line 384: "Figure 1" is incorrectly labeled (should be "Figure 2", as it describes aquatic organisms, while Figure 1 focuses on mammals). Verify all figure cross-references (e.g., Line 384’s "Figure 1" → "Figure 2") to match the actual figure content.
Author Response
Comments 1: Resolve inconsistencies in nanoplastic size definitions and table annotations to enhance precision. The manuscript initially defines NPs as 1 nm–1 μm (Line 41–42), aligning with metabolomics studies. However, in Table 2 (e.g., PS-NPs with size 50–100 nm, PET-NPs with 700±300 nm), some "nanoplastics" exceed 1 μm (e.g., 700±300 nm overlaps with microplastics). This inconsistency confuses readers and undermines the review’s precision. Recommendation: - Add a footnote to Table 2 clarifying that particles exceeding 1 μm are included due to their classification in cited studies, while reaffirming the manuscript’s 1 nm–1 μm definition. - In the Introduction (Section 1), explicitly acknowledge that size classifications vary across studies and explain why the 1 nm–1 μm range is prioritized (e.g., consistency with metabolomic research).
Response 1:
Thank you for this valuable comment regarding the inconsistency in nanoplastic size definitions. As the reviewer noted, the manuscript adopts the 1–1000 nm range for “nanoplastics,” which aligns with classifications commonly used in metabolomics-based research. To clarify this point, we have added the following explanatory sentence to the Introduction(lines 48-52):
“Because the size definition of nanoplastics differs among studies, the particle-size ranges applied in the literature are not always consistent. In this review, we adopt the 1–1000 nm range for the term ‘nanoplastics,’ as this classification aligns with the definitions most commonly used in metabolomics-based research.”
Although the reviewer referred to Table 2, we confirmed that all entries in Table 2 fall within the 1 μm range. Instead, Table 1 contains particle sizes exceeding 1 μm as reported in the original cited studies. Therefore, we added the clarifying footnote to Table 1, rather than Table 2, to accurately address the reviewer’s concern.
The footnote added to Table 1 is as follows:
“Size definitions of nanoplastics vary across studies. While this review prioritizes 1–1000 nm, some cited papers classify particles slightly exceeding 1 μm as ‘nano-plastics’; we retain the original authors’ terminology here to reflect the source data”
Comments 2: Strengthen cross-species comparative analysis by adding a dedicated subsection and visual summary: The manuscript separately discusses mammalian and aquatic systems but lacks a dedicated section comparing shared vs. distinct metabolic disruption patterns. For example, both systems exhibit lipid metabolism disturbance, but aquatic organisms (e.g., fish gills) show unique mucus metabolism alterations that are not contrasted with mammalian barriers (e.g., intestinal epithelium). Recommendation: - Add a new subsection (e.g., "3.4 Comparative Metabolic Signatures Across Mammalian and Aquatic Systems") to synthesize: - Shared pathways (e.g., arachidonic acid metabolism, TCA cycle impairment). - Species-specific differences (e.g., mucus metabolism in aquatic organisms vs. gut-liver axis disruption in mammals). - Use a concise table (e.g., Table 3) to list key perturbed pathways, organs, and species, enhancing visual comparison.
Response 2:
We sincerely thank the reviewer for this insightful suggestion. In accordance with the recommendation, we have substantially strengthened the cross-species comparative analysis by adding an entirely new subsection and a visual summary table.
- A new subsection has been added (starting at line 675):
- Section 4. Comparative metabolic signatures across mammals and aquatic organisms
This subsection synthesizes the major metabolic pathways commonly affected by nanoplastics in both mammalian and aquatic models, including shared disruptions in the TCA cycle and arachidonic acid metabolism, while also highlighting species-specific alterations such as glycerophospholipid metabolism in aquatic organisms and steroid-related metabolism in mammals. - Organ-level comparison has been added (c):
- Section 4.1 Common metabolic pathways by organs (lines 693-710)
This subsection compares organ-specific metabolic alterations (liver, intestine, brain) and provides interpretive commentary on analogous tissues (lungs–gills and fetus–larvae). - A new comparative table (Table 3, line 711) has been added:
- Table 3. Comparative metabolic studies of key pathway alterations due to nanoplastic toxicity in mammals and aquatic organisms
This table summarizes the major perturbed pathways by organ across the two system types and highlights overlapping pathways in bold, enabling clear visualization of cross-species similarities and differences.
These additions directly address the reviewer’s request and significantly improve cross-species integrative interpretation within the manuscript.
Revisions made at:
- Pages 26, Section 4 and Section 4.1
- Page 27, Table 3
Comments 3: Ambiguous Description of Mechanistic Links: The manuscript repeatedly mentions that NPs induce oxidative stress and mitochondrial dysfunction, but the mechanistic link between these processes and specific metabolic changes is unclear. Recommendation: - For key pathways (e.g., TCA cycle, lipid peroxidation), add a brief mechanistic explanation using cited studies. Use a simple schematic (e.g., Figure 3) to illustrate the cascade: NP exposure → ROS/mROS → mitochondrial dysfunction → metabolic pathway disruption.
Response 3:
We sincerely thank the reviewer for this insightful comment. In response, we have substantially revised the manuscript to clarify the mechanistic links between oxidative stress, mitochondrial impairment, and the key metabolic pathways disrupted by nanoplastic exposure.
- A new subsection titled “4.2 Linkage between nanoplastic toxicity mechanisms and key pathway alterations” has been added (starting at line 715).
This section provides detailed mechanistic explanations of how nanoplastics disrupt - glycerophospholipid metabolism,
- arachidonic acid metabolism, and
- the tricarboxylic acid (TCA) cycle.
These explanations incorporate relevant studies demonstrating how ROS/mROS generation and mitochondrial dysfunction drive downstream metabolic disturbances.
(Revised manuscript: Section 4.2, Lines 714-776) - A new schematic diagram (Figure 3) has been added, illustrating the mechanistic cascade:
Nanoplastic exposure → ROS/mROS generation → mitochondrial impairment → metabolic pathway alterations → toxicological outcomes.
(Revised manuscript: Figure 3, line 777)
These revisions directly address the reviewer’s concerns and now provide a clear mechanistic framework supporting the metabolic alterations described throughout the manuscript.
Comments 4: Typographical Errors: Line 384: "Figure 1" is incorrectly labeled (should be "Figure 2", as it describes aquatic organisms, while Figure 1 focuses on mammals). Verify all figure cross-references (e.g., Line 384’s "Figure 1" → "Figure 2") to match the actual figure content.
Response 4:
We thank the reviewer for pointing out this typographical issue. We have corrected the mislabeled figure reference on Line 384 from “Figure 1” to “Figure 2”. In addition, we carefully re-checked all figure citations throughout the manuscript and ensured that every in-text reference now matches the corresponding figure content and numbering.
Revisions made in the revised manuscript: Line 447 (Figure 1 → Figure 2), and all figure cross-references have been verified and corrected where necessary.

Reviewer 2 Report
Comments and Suggestions for Authors
The authors summarize recent metabolomic studies on nanoplastics, showing how polystyrene and PET disrupt metabolism in mammals and aquatic organisms. Key affected pathways include lipid, energy, and amino acid metabolism, with changes observed in organs like the liver, intestines, gills, and brain, often alongside oxidative stress and inflammation. They highlight metabolomics as a valuable tool for understanding nanoplastic toxicity and stress the need to combine it with other methods for a comprehensive health risk assessment.
Major comments:
Once an abbreviation for a term is introduced, it should be used consistently throughout the entire text – for example page 8, line 200, abbreviation ROS.
Regarding presenting literature results, several relevant references are missing; it is recommended to include them to support the statements. For example, page 5, line 180, In AOM/DSS-treated mice….; page 8, line 333, In mouse models….
When presenting results, it is essential to specify the exact model system and the methodology used for metabolomic analysis, as different methods can yield different outcomes.
On page 8, line 355, it is stated that changes occur in the brain, but no details are provided regarding the type of changes or the model system used. A more precise description of nanoplastic effects on the mammalian brain should be included, similar to the findings reported in aquatic organisms.
It is also necessary to outline some of the mechanisms leading to the observed damage, as the current description is too general and the changes are only listed without context. Additionally, one of the mechanisms involving NF-κB is mentioned in the conclusion, where it does not belong, and should instead be discussed in the main text.
The conclusion should summarize the results rather than restating the observed changes or proposing a mechanism.
One of the main criticisms is related to the limitations of the study, in the sentence under the second limitation. It was written that “future studies should incorporate metabolomic analyses to better elucidate the biochemical mechanisms of nanoplastic toxicity. Most studies, as mentioned above, were performed by transcriptomics, … In fact, the entire text, starting from the title, focuses on metabolomics.
Author Response
Comments 1: Once an abbreviation for a term is introduced, it should be used consistently throughout the entire text – for example page 8, line 200, abbreviation ROS.
Response 1:
We appreciate the reviewer’s observation regarding the inconsistent use of abbreviations. In response, we carefully reviewed the entire manuscript to ensure that each abbreviation is introduced only once in its full form and subsequently used consistently throughout the text. The specific instance identified by the reviewer has been corrected (now line 204), where “reactive oxygen species (ROS)” is introduced properly and ROS is used in all subsequent occurrences. In addition, we performed a manuscript-wide check to ensure consistent use of other abbreviations, including ATP, ETC, GP, AA, and TCA, and revised any remaining inconsistencies. These corrections enhance clarity and stylistic uniformity across the manuscript.
Comments 2: Regarding presenting literature results, several relevant references are missing; it is recommended to include them to support the statements. For example, page 5, line 180, In AOM/DSS-treated mice….; page 8, line 333, In mouse models….
Response 2:
We appreciate the reviewer’s helpful suggestion. In response, we carefully reviewed all literature-derived statements throughout the manuscript and added the appropriate references wherever they were missing. The following missing citations have now been incorporated into the revised text:
Line 185: Added the appropriate reference supporting the reported metabolic alterations in AOM/DSS-treated mouse models.
Line 192: Inserted the original reference describing ABC transporter enrichment, alkaloid/phenylpropanoid biosynthesis changes, bile secretion disruption, and reduced mucus production following PS-NP exposure.
Line 380: Added the reference for metabolic changes in HKC cells, including disruption of energy and nucleotide metabolism.
Line 385: Added the reference documenting alterations in serine–glycine conversion, fatty acid metabolism, and GSH-related responses.
Line 390: Added the reference supporting increased purine/pyrimidine metabolites and elevated uric acid after PS-NP exposure.
In addition to these specific corrections, we conducted a manuscript-wide review to ensure that all statements summarizing previous findings—particularly in Sections 2 and 3—are now consistently supported by the appropriate references. Reference numbering has been updated accordingly.
Comments 3: When presenting results, it is essential to specify the exact model system and the methodology used for metabolomic analysis, as different methods can yield different outcomes.
Response 3:
Thank you for pointing out the importance of clearly specifying the model systems and analytical methodologies used in the cited metabolomic studies. We fully agree that metabolomic outcomes can vary substantially depending on the organism, tissue type, particle characteristics, and analytical platform used.
In response, we have thoroughly revised the manuscript to explicitly describe both (1) the model system and (2) the metabolomic method for each cited study throughout the text. These details have now been systematically incorporated at all relevant points.
- Reporting standards for metabolomic analytical methods were clarified
- Added generalized methodological standards in
Page 4, Line 151 and Page 15, Line 444.
- Specific metabolomic methodologies used in each referenced study were added
These descriptions now appear in:
- Page 5: Lines 184, 187
- Page 6: Lines 220, 233
- Page 7: Line 267
- Page 8: Lines 339, 355
- Page 9: Lines 396
- Page 17: Lines 499, 521, 530
- Page 18: Lines 559, 593
- Page 19: Lines 622, 625, 646
These additions specify whether the study used LC–MS, GC–MS, NMR-based metabolomics, targeted vs. untargeted workflows, and the analytical platform used.
- Details of experimental model systems were specified
- Model organism, tissue/cell type, and exposure context were clarified in
Page 5, Line 174, and additional locations where necessary.
Comments 4: On page 8, line 355, it is stated that changes occur in the brain, but no details are provided regarding the type of changes or the model system used. A more precise description of nanoplastic effects on the mammalian brain should be included, similar to the findings reported in aquatic organisms.
Rsponse 4:
Thank you for this valuable comment. As suggested, we substantially revised the section describing nanoplastic-induced effects on the mammalian brain (now Section 2.1.5, lines 275-326). The previous version provided only a general statement, but the revised text now includes detailed descriptions of the specific metabolic alterations and the model systems from which they were derived. We incorporated metabolomic findings from mouse frontal lobe tissue and SH-SY5Y neuronal cells, including disruptions in amino acid metabolism, kynurenine pathway intermediates, glycerolipid and steroid biosynthesis, neurotransmitter-related metabolites, and cytochrome P450–regulated xenobiotic metabolism. In addition, we expanded the mechanistic context by describing BBB penetration, microglial activation, and mitochondrial dysfunction supported by NMR- and MS-based metabolomic analyses. To improve clarity and narrative flow, we also added a brief introductory sentence to position the brain as a metabolically vulnerable organ and included a closing transition linking these findings to systemic toxicity. These revisions provide a more precise, comprehensive, and model-specific description of nanoplastic-induced metabolic disturbances in the mammalian brain, fully addressing the reviewer’s concern.
Comments 5: It is also necessary to outline some of the mechanisms leading to the observed damage, as the current description is too general and the changes are only listed without context. Additionally, one of the mechanisms involving NF-κB is mentioned in the conclusion, where it does not belong, and should instead be discussed in the main text.
Response 5:
Thank you for this valuable comment. We agree that the previous version listed metabolic alterations without providing sufficient mechanistic context, and that the mention of NF-κB in the conclusion was inappropriate. To fully address this issue, we have substantially revised the manuscript by adding a new dedicated subsection titled “4.2 Linkage between nanoplastic toxicity mechanisms and key pathway alterations” (starting at line 715). This new section synthesizes mechanistic insights from the literature and explains how nanoplastic exposure initiates a cascade of cellular events—beginning with ROS/mROS generation, oxidative stress, and mitochondrial impairment—that subsequently drives disruptions in glycerophospholipid metabolism, arachidonic acid signaling, and the TCA cycle. Each pathway is now described with mechanistic detail, including how changes in membrane lipid remodeling alter receptor and ion-channel function, how arachidonic-acid–derived eicosanoids mediate inflammation, and how mitochondrial dysfunction reduces ATP production and redox capacity, ultimately contributing to neurotoxicity, developmental defects, and impaired energy homeostasis. Importantly, we relocated the previously mentioned NF-κB-related mechanism from the conclusion to this mechanistic section, where it is now discussed within the appropriate biological framework. These revisions provide a much clearer, mechanistically integrated explanation of how nanoplastic exposure leads to the observed metabolic disturbances, fully addressing the reviewer’s concern and strengthening the scientific coherence of the manuscript.
Comments 6: The conclusion should summarize the results rather than restating the observed changes or proposing a mechanism.
Response 6:
Thank you for this valuable comment. We agree that the Conclusion section should serve as a concise synthesis rather than reiterating detailed pathway alterations or introducing mechanistic explanations. In response, we have substantially revised the Conclusion so that it now focuses only on the overarching findings and future research needs.
Importantly, the mechanistic content that was previously included in the Conclusion has been relocated to the main body of the manuscript. Specifically, we created a new subsection (Section 4.2: Linkage between nanoplastic toxicity mechanisms and key pathway alterations) and integrated the mechanistic discussion into the appropriate sections of the manuscript (Sections 2–4). This ensures that mechanistic interpretation is provided within the scientific context of the main text rather than in the concluding remarks.
The revised Conclusion now provides a concise summary aligned with the reviewer’s recommendation. These changes can be found in Lines 715–776 (new Section 4.2) and Lines 781–806 (revised Conclusion) of the updated manuscript.
Comments 7: One of the main criticisms is related to the limitations of the study, in the sentence under the second limitation. It was written that “future studies should incorporate metabolomic analyses to better elucidate the biochemical mechanisms of nanoplastic toxicity. Most studies, as mentioned above, were performed by transcriptomics, … In fact, the entire text, starting from the title, focuses on metabolomics.
Response 7:
We thank the reviewer for this helpful observation. In the revised manuscript, we modified the second limitation to remove any implication that metabolomics has not yet been applied in nanoplastic research. Instead, the revised text clarifies that metabolomics is already a central analytical approach in this field but remains limited in depth and scope compared with transcriptomics. To address the reviewer’s concern, we rewrote the limitation to emphasize the need for expanded and more integrative metabolomics applications—particularly within multi-omics frameworks—rather than “introducing” metabolomics. The revised conclusion now states(lines 793-798):
“Secondly, although metabolomics already plays a central role in this field, its application remains limited in depth and scope compared with transcriptomics. Future studies should expand metabolomics-based analyses—particularly within multi-omics frame-works—to more comprehensively elucidate the biochemical mechanisms of nanoplastic toxicity. Such integration will help clarify how gene-expression changes translate into metabolic outcomes and toxic phenotypes.”
This revised wording aligns with the reviewer’s suggestion and removes the inconsistency present in the earlier version.

Round 2
Reviewer 1 Report
Comments and Suggestions for Authors
Some comments and suggestions major revision are listed below.
1. Most cited studies focus on polystyrene nanoplastics, leading to insufficient coverage of other environmentally relevant polymer types (e.g., PVC, PP); expanding the scope to include diverse nanoplastics will improve the generalizability of conclusions.
2. The section on "nanoplastic toxicity in aquatic invertebrates" (Section 3.2) is underrepresented. Only 2 studies (Daphnia magna, Cherax quadricarinatus) are cited, and there is a lack of data on ecologically critical groups such as bivalves and zooplankton. It is recommended to supplement 3-5 recent (2023-2025) metabolomic studies on aquatic invertebrates to improve the coverage of aquatic ecosystemsand balance the cross-species analysis.
3. Some key claims lack direct data support. For example, in Section 2.1.7, it states that "nanoplastics affect the vascular system by inducing hemolysis," but no specific metabolomic indicators (e.g., changes in hemoglobin metabolites, lipid peroxidation markers) or corresponding literature citations are provided. It is suggested to add specific metabolite data and reference sources to enhance persuasiveness.
4. The discussion on the mechanistic links between metabolic disruptions and toxic phenotypes could be strengthened by integrating more molecular signaling pathways (e.g., MAPK, NF-κB) with metabolomic changes.The mechanistic link between "glycerophospholipid metabolism disruption" and "membrane function impairment" is not sufficiently elaborated. For example, it mentions that glycerophospholipid changes affect membrane fluidity but does not specify which metabolites (e.g., phosphatidylcholine, phosphatidylethanolamine) are key regulators or how their content changes alter membrane properties. It is recommended to add a simplified metabolic pathway diagram (supplementing Figure 3) to show the specific molecular mechanisms.
5. The discussion on "translational value" is insufficient. It fails to explain how metabolomic findings (e.g., disturbed glycerophospholipid metabolism in the liver) can be applied to specific scenarios such as environmental monitoring (e.g., developing metabolite-based detection kits) or human health risk assessment (e.g., setting safe exposure limits). It is recommended to add a "translational application" subsection to enhance practical guiding significance.
6. The limitations of metabolomics itself are not discussed. For example, metabolomics is highly sensitive to experimental conditions (e.g., sample storage, detection platform), which may lead to inconsistent results across studies. The review should acknowledge these technical limitations and propose solutions (e.g., standardizing sample processing protocols) to provide a more objective perspective.
7. The conclusion part merely summarizes existing findings without proposing prospective directions for future metabolomic research (e.g., multi-omics integration with microbiomics, real environmental nanoplastics exposure models). Supplementing forward-looking content will enhance the guiding value of the review.
Comments on the Quality of English Language
The English language quality of the paper is highly competent and consistent with academic writing standards, demonstrating clarity, precision, and appropriate technical expression for a peer-reviewed journal in molecular sciences.
Author Response
Comment 1: Most cited studies focus on polystyrene nanoplastics, leading to insufficient coverage of other environmentally relevant polymer types (e.g., PVC, PP); expanding the scope to include diverse nanoplastics will improve the generalizability of conclusions.
Response 1: We sincerely thank the reviewer for the valuable suggestion regarding the inclusion of metabolomics studies involving polymer types other than polystyrene. We fully agree that expanding the diversity of polymer types is important for improving the generalizability of conclusions in the field of nanoplastic toxicology.
To address this point, we would like to clarify that our Tables 1 and 2 already include the available metabolomics studies involving PVC, PP, PC, PE, and PET, in addition to PS. However, during our comprehensive literature search, we found that metabolomics-based nanoplastic research beyond polystyrene remains extremely limited. This appears to reflect the current developmental stage of the field rather than an omission in our manuscript.
Historically, nanoplastic toxicology research has evolved in a stepwise manner:
(1) initial mechanistic toxicology studies using PS as an experimental model,
(2) subsequent in vitro and in vivo characterization of PS toxicity across biological systems,
(3) later expansion into transcriptomic analyses, and
(4) only in recent years, the application of metabolomics.
As a result, most metabolomics-based studies currently available in the literature are naturally centered on PS-NPs, whose toxicological profiles have been most extensively characterized.
During our search, we identified several studies involving non-PS polymers (e.g., PVC, PP, PET). However, these publications generally used microplastics rather than nanoplastics, with particle sizes far exceeding nanoplastic criteria. Importantly, these were not borderline cases slightly above 1 μm; instead, the particles ranged from 5 μm to over 20 μm, clearly placing them in the microplastic domain. In several studies, the physicochemical characterization of particle size was also incomplete or imprecise. Because our review specifically focuses on nanoplastics within the 1 nm–1 μm range combined with metabolomic analysis, these studies did not meet the inclusion criteria and were therefore excluded. Representative examples include:
PVC-related (microplastic scale)
- Isolated and combined toxicity of PVC microplastics and copper on Pinctada fucata martensii: Immune, oxidative, and metabolomics insights
- Polyvinyl chloride microplastics induced gut barrier dysfunction, microbiota dysbiosis and metabolism disorder in adult mice
- Multi-Omics Analysis Reveals the Toxicity of Polyvinyl Chloride Microplastics toward BEAS-2B Cells
PP-related (microplastic scale)
- Dose-Dependent Cytotoxicity of Polypropylene Microplastics (PP-MPs) in Two Freshwater Fishes
- Impacts of polypropylene microplastics on lipid profiles of mouse liver uncovered by lipidomics analysis and Raman spectroscopy
PET-related (microplastic scale)
- Exposure to polyethylene terephthalate micro(nano)plastics exacerbates inflammation and fibrosis after myocardial infarction by reprogramming the gut and lung microbiota and metabolome
Mixed microplastics
- In Vivo Tissue Distribution of Microplastics and Systemic Metabolomic Alterations After Gastrointestinal Exposure
Given the very limited availability of metabolomics studies using true nanoplastics of diverse polymer types, the predominance of PS-NP studies in our review reflects the current state of published research. Nonetheless, we fully agree with the reviewer that future studies should expand toward a broader diversity of nanoplastic polymers. This point has now been explicitly discussed in the conclusion and perspectives section of the revised manuscript..
We hope this clarification addresses the reviewer’s concern, and we appreciate the opportunity to strengthen the manuscript based on this thoughtful comment.
Comment 2: The section on "nanoplastic toxicity in aquatic invertebrates" (Section 3.2) is underrepresented. Only 2 studies (Daphnia magna, Cherax quadricarinatus) are cited, and there is a lack of data on ecologically critical groups such as bivalves and zooplankton. It is recommended to supplement 3-5 recent (2023-2025) metabolomic studies on aquatic invertebrates to improve the coverage of aquatic ecosystemsand balance the cross-species analysis.
Response 2: We thank the reviewer for emphasizing the ecological importance of aquatic invertebrates and for suggesting that this section be strengthened. As noted in our original manuscript, nanoplastic–metabolomics studies in aquatic invertebrates remain extremely limited, especially compared with vertebrate models. The scarcity of such studies—rather than omission—reflects a genuine gap in the current literature.
Nevertheless, in response to the reviewer’s recommendation, we conducted an additional comprehensive literature search (2023–2025) and successfully identified three new nanoplastic–metabolomics studies involving ecologically relevant invertebrate taxa. These studies have now been incorporated into Section 3.2 (lines 678-687):
- Marine rotifer (Brachionus plicatilis) exposed to PS-NPs [136]
– Altered pathways: alanine, aspartate and glutamate metabolism; TCA cycle; purine metabolism; pyrimidine metabolism. - Triploid Fujian oyster (Crassostrea angulata) exposed to PS-NPs [137]
– Altered pathways: arachidonic acid metabolism; arginine and proline metabolism; glutathione metabolism; lysine degradation; phenylalanine/tyrosine/tryptophan biosynthesis; fructose/mannose metabolism; ubiquinone and other terpenoid–quinone biosynthesis; thiamine metabolism; purine metabolism. - Marine clam (Paphia undulata) analyzed by GC-TOF/MS following PS-NP exposure [138]
– Altered pathways: phenylalanine/tyrosine/tryptophan biosynthesis; galactose metabolism; starch and sucrose metabolism.
These three studies have now been fully integrated into the main text of Section 3.2, expanding the representation of aquatic invertebrates to include zooplankton (rotifers), bivalves (oysters), and marine clams.
Additionally, we updated Table 2 to include these three newly added studies, ensuring that the tabulated summary of nanoplastic–metabolomics literature now reflects all available invertebrate findings (pages 25-26).
Finally, we have strengthened the limitations and future directions sections to explicitly state that nanoplastic + metabolomics research in aquatic invertebrates is still in its early stages, and that the limited number of available studies highlights an urgent need for further work.
We believe these revisions substantially enhance the ecological coverage and cross-species balance of the review while faithfully representing the current state of the scientific literature.
Comment 3: Some key claims lack direct data support. For example, in Section 2.1.7, it states that "nanoplastics affect the vascular system by inducing hemolysis," but no specific metabolomic indicators (e.g., changes in hemoglobin metabolites, lipid peroxidation markers) or corresponding literature citations are provided. It is suggested to add specific metabolite data and reference sources to enhance persuasiveness.
Response 3: We thank the reviewer for this important comment. In the revised manuscript, we have substantially expanded Section 2.1.7 to include both mechanistic evidence for hemolysis and specific metabolomic data supporting nanoplastic-induced vascular and hematopoietic effects. (lines 377-395)
First, we added detailed mechanistic descriptions of hemolysis, showing that PS-NPs increase intracellular Ca²⁺, deplete ATP and GSH, activate erythrocyte scramblase, disrupt membrane phospholipid asymmetry, and promote phosphatidylserine externalization and microvesicle formation. These mechanisms are now supported with proper citations.
Second, in accordance with the reviewer’s suggestion, we clarified that no metabolomics studies have yet profiled hemoglobin-derived metabolites or lipid peroxidation markers in erythrocytes, and we explicitly state this limitation.
Third, to strengthen metabolomic evidence in hematopoietic tissues, we incorporated a recent study showing that PS-NP exposure alters ten metabolites in hematopoietic stem/progenitor cells (HSPCs) and enriches multiple metabolic pathways, including the TCA cycle, D-glutamine and D-glutamate metabolism, alanine–aspartate–glutamate metabolism, lysine biosynthesis, vitamin B6 metabolism, propanoate metabolism, and butanoate metabolism. These findings provide direct metabolomic support for NP-induced hematopoietic disruption.
The revised section now provides a comprehensive mechanistic and metabolomic basis for nanoplastic-induced hematologic and vascular toxicity, thereby addressing the reviewer’s concern fully.
Comment 4: The discussion on the mechanistic links between metabolic disruptions and toxic phenotypes could be strengthened by integrating more molecular signaling pathways (e.g., MAPK, NF-κB) with metabolomic changes.The mechanistic link between "glycerophospholipid metabolism disruption" and "membrane function impairment" is not sufficiently elaborated. For example, it mentions that glycerophospholipid changes affect membrane fluidity but does not specify which metabolites (e.g., phosphatidylcholine, phosphatidylethanolamine) are key regulators or how their content changes alter membrane properties. It is recommended to add a simplified metabolic pathway diagram (supplementing Figure 3) to show the specific molecular mechanisms.
Response 4: We sincerely thank the reviewer for this insightful and constructive recommendation. In response, we have substantially strengthened Section 4.2 by adding two dedicated mechanistic subsections that directly address the reviewer’s points:
- Mechanistic link between glycerophospholipid metabolism and membrane function (lines 774-799)
We incorporated a detailed explanation describing which specific glycerophospholipid species (e.g., PC 16:0, PC 18:0, LysoPC 18:0, PC 18:1, AA, LA, DHA) are disrupted following nanoplastic exposure and how these compositional shifts impair membrane fluidity, bilayer packing, mitochondrial membrane function, and synaptic activity.
This new paragraph clarifies how SFA-enriched phosphatidylcholines increase membrane rigidity, whereas reduced PUFA-containing phospholipids decrease membrane fluidity, ultimately compromising cellular and mitochondrial functionality.
These details directly address the reviewer’s request to specify “which metabolites” and “how their content changes alter membrane properties.”
Importantly, we also incorporated the reviewer’s request to elaborate on regulatory lipid classes by adding a new explanation of the PC/PE ratio as a key determinant of membrane curvature, integrity, and permeability. The revised paragraph clarifies that excessive PE enrichment induces negative curvature stress and barrier instability, whereas excessive PC enrichment leads to overly rigid membranes that impair vesicle trafficking and mitochondrial respiration. These details directly address the reviewer’s question regarding “which metabolites” and “how their content changes alter membrane properties.”
- Integration of molecular signaling pathways with metabolic disruptions (lines 847-869)
We added a second paragraph clarifying the upstream toxicological signaling cascade:
nanoplastic exposure → ROS accumulation → activation of MAPK (ERK, JNK, p38) → NF-κB nuclear translocation → PLA2 activation → AA release and COX-2 upregulation → eicosanoid overproduction → oxidative & inflammatory metabolic disruption. This new section explicitly links oxidative stress, MAPK/NF-κB signaling, and lipid/energy metabolic pathways, as requested.
- Finally, we supplemented Figure 3 by adding a new module illustrating the mechanistic flow from glycerophospholipid alterations to membrane-function impairment, including changes in lipid saturation, PC/PE ratio imbalance, and resulting effects on membrane rigidity and cellular stress. This visual addition complements the revised text and provides readers with a concise schematic overview of the membrane-level mechanisms associated with glycerophospholipid disruption.
We believe these revisions fully address the reviewer’s constructive suggestions and significantly enhance the mechanistic clarity of Section 4.2.
Comment 5: The discussion on "translational value" is insufficient. It fails to explain how metabolomic findings (e.g., disturbed glycerophospholipid metabolism in the liver) can be applied to specific scenarios such as environmental monitoring (e.g., developing metabolite-based detection kits) or human health risk assessment (e.g., setting safe exposure limits). It is recommended to add a "translational application" subsection to enhance practical guiding significance.
Response 5: We appreciate the reviewer’s insightful comment regarding the need to further elaborate on the translational value of metabolomic findings. To address this, we have added a fully new subsection titled “4.3 Translational Applications of Metabolomic Findings in Nanoplastic Toxicology” in the revised manuscript (lines 875-901).
In this section, we now provide a clearer explanation of how metabolomics can be applied beyond mechanistic understanding, with direct relevance to environmental monitoring and human health risk assessment. Specifically, we discuss how recurrent metabolic alterations—such as disruptions in glycerophospholipid metabolism, TCA cycle imbalance, and oxidative stress–related amino acid metabolism—may serve as candidate biomarkers of nanoplastic exposure. We highlight how such biomarkers could eventually be incorporated into targeted metabolite-based panels or field-deployable assays for pollution surveillance, early-warning systems, and human exposure assessment.
We also elaborate on how dose-dependent metabolic responses can help establish biologically relevant thresholds useful for risk evaluation or for informing future regulatory exposure limits. In addition, we describe how species-specific metabolic fingerprints can complement existing ecotoxicological indices and provide more sensitive tools for evaluating ecological impacts.
Finally, the added subsection emphasizes that while such translational applications remain preliminary due to limited datasets and the early stage of nanoplastic–metabolomics research, metabolomics holds substantial potential to bridge mechanistic toxicology with practical monitoring and regulatory frameworks.
These additions now appear in Section 4.3 of the revised manuscript, substantially strengthening the translational perspective as recommended.
Comment 6: The limitations of metabolomics itself are not discussed. For example, metabolomics is highly sensitive to experimental conditions (e.g., sample storage, detection platform), which may lead to inconsistent results across studies. The review should acknowledge these technical limitations and propose solutions (e.g., standardizing sample processing protocols) to provide a more objective perspective.
Response 6: We thank the reviewer for this important comment.
In the revised manuscript, we have now added a dedicated discussion of the methodological limitations inherent to metabolomics and proposed corresponding solutions (Introduction, lines 123-133).
Specifically, we now state:
“Although metabolomics offers clear advantages for characterizing biochemical alterations, several methodological limitations must also be acknowledged. First, metabolite levels are highly sensitive to pre-analytical and analytical conditions, including sampling timing, storage procedures, extraction methods, and variability across analytical platforms (e.g., NMR, LC–MS, GC–MS). Such factors can introduce inter-study variability, particularly in the absence of standardized protocols [38]. Second, accurate interpretation of metabolomic profiles requires detailed knowledge of metabolite chemistry, biological function, and pathway context within each organism, which can complicate cross-study comparisons [38]. Nevertheless, recent efforts toward protocol harmonization, quality-control stand-ardization, and cross-platform validation provide a strong foundation for improving re-producibility and reliability in future metabolomics research [39-41].”
Comment 7: The conclusion part merely summarizes existing findings without proposing prospective directions for future metabolomic research (e.g., multi-omics integration with microbiomics, real environmental nanoplastics exposure models). Supplementing forward-looking content will enhance the guiding value of the review.
Response 7: We sincerely thank the reviewer for this insightful recommendation.
In accordance with the reviewer’s suggestion, we have substantially revised and expanded the conclusion section to incorporate a clearer and more forward-looking perspective. The revised conclusion now includes:
- Integration of multi-omics approaches (lines 924-930)
We added a comprehensive discussion on integrating metabolomics with transcriptomics, lipidomics, proteomics, and microbiomics to clarify how molecular signaling pathways (e.g., oxidative stress responses, inflammatory signaling, mitochondrial dysfunction) give rise to downstream metabolic alterations.
This directly addresses the reviewer’s request for mechanistic and multi-layered future directions.
- Emphasis on real-environment nanoplastic exposure models (lines 930-935)
We introduced new text explaining the need for weathered particles, mixed-polymer nanoplastics, and biofilm-coated particles, which better reflect actual environmental conditions compared to pristine PS-NPs.
This responds to the reviewer’s comment on adopting realistic exposure scenarios.
- Translational applications of metabolomic findings (lines 936-940)
We added a dedicated explanation of how metabolomic biomarkers can support:
- environmental monitoring and ecosystem early-warning systems,
- human health risk evaluation and susceptibility assessment.
This new content strengthens the practical relevance of metabolomics, as suggested by the reviewer.
- Methodological standardization (940-946)
We incorporated the reviewer’s recommendation to discuss the importance of standardizing metabolomics workflows, including sample handling, storage, platform harmonization, and quality-control procedures.
This addition highlights the technical requirements for reliable cross-study comparison.
- Final integrated conclusion (947-951)
The entire conclusion was then re-organized to provide a unified, forward-looking perspective while maintaining a clear and concise closing statement.

Reviewer 2 Report
Comments and Suggestions for Authors
The authors have satisfactorily addressed my comments.
Author Response
The authors have satisfactorily addressed my comments.
Round 3
Reviewer 1 Report
Comments and Suggestions for Authors
The author responded to each of the reviewers' comments one by one and made numerous revisions and additions.
Comments on the Quality of English LanguageThe English language quality of the paper is highly competent and consistent with academic writing standards, demonstrating clarity, precision, and appropriate technical expression for a peer-reviewed journal in molecular sciences.